# When to Ask for Help: Proactive Interventions in Autonomous Reinforcement Learning

**Annie Xie,**\* **Fahim Tajwar,**\* **Archit Sharma,**\* **Chelsea Finn**
Department of Computer Science
Stanford University
{anniexie,tajwar93,architsh,cbfinn}@stanford.edu

## Abstract

A long-term goal of reinforcement learning is to design agents that can autonomously interact and learn in the world. A critical challenge to such autonomy is the presence of irreversible states which require external assistance to recover from, such as when a robot arm has pushed an object off of a table. While standard agents require constant monitoring to decide when to intervene, we aim to design proactive agents that can request human intervention only when needed. To this end, we propose an algorithm that efficiently learns to detect and avoid states that are irreversible, and proactively asks for help in case the agent does enter them. On a suite of continuous control environments with unknown irreversible states, we find that our algorithm exhibits better sample- and intervention-efficiency compared to existing methods. Our code is publicly available at https://sites.google.com/view/proactive-interventions.

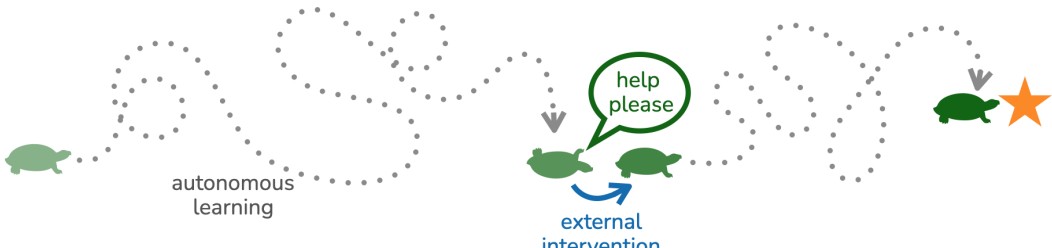

Figure 1: Autonomous agents struggle to make progress without external interventions when they are stuck in an irreversible state. Reinforcement learning agents therefore need active monitoring throughout training to detect and intervene when the agent reaches an irreversible state. Enabling the agents to detect irreversible states and proactively request for help can substantially reduce the human monitoring required for training agents.

## 1 Introduction

A reinforcement learning (RL) agent should be able to autonomously learn behavior by exploring in and interacting with its environment. However, in most realistic learning environments, there are irreversible states from which the agent cannot recover on its own. For example, a robot arm can inadvertently push an object off the table, such that an external supervisor must return it back to the robot's workspace to continue the learning process. Current agents demand constant monitoring to decide when the agent enters an irreversible state and therefore when to intervene. In this work, we aim to build greater autonomy into RL agents by addressing this problem. In particular, we envision

---

\*equal contribution

36th Conference on Neural Information Processing Systems (NeurIPS 2022).

proactive agents that can instead detect irreversible states, proactively request interventions when needed, and otherwise learn autonomously.

Prior works have studied autonomy in RL, aiming to minimize the number of human-provided resets at the end of each episode, but generally assume the environment is fully reversible [19, 53, 35, 34, 17]. Our work focuses on settings with potential irreversible states and algorithms to avoid such states. A related desiderata, however, arises in the safe RL setting; safe RL methods aim to learn policies that minimize visits to unsafe states, and the developed approaches are designed to avoid those particular parts of the state space [1, 10, 39, 37, 43, 41]. Prior safe RL algorithms assume that the agent is given a safety label on demand for *every* state it visits. In contrast, an autonomous agent may not know when it has reached an irreversible state (such as knocking an important object off a table), and an algorithm in this setting should instead learn to both detect and avoid such states, while minimizing queries about whether a state is reversible.

With this in mind, we design our setup to provide help to the agent in two ways: through an environment reset or through the reversibility label of a state. However, unlike in safe RL, we can reduce the labeling requirement with a simple observation: all states *proceeding* an irreversible state are irreversible, and all states *preceding* a reversible state will be reversible. Based on this observation, we design a scheme based on binary search to generate reversibility labels for a trajectory of length $T$ using at most $\mathcal{O}(\log T)$ label queries, compared to the $\mathcal{O}(T)$ queries made by safe RL methods. We further reduce labeling burden by only querying labels in a large batch at the end of each extended episode, i.e. typically only after tens of thousands of steps. By combining this label efficient scheme with proactive requests for an intervention and batch of labels, we can enable agents to learn amidst irreversible states with a high degree of autonomy.

Concretely, we propose a framework for reversibility-aware autonomous RL, which we call *proactive agent interventions (PAINT)*, that aims to minimize human monitoring and supervision required throughout training. First, we train a reversibility-aware $Q$-value function that penalizes visits to irreversible states. Second, the reversibility labels are generated by a label-efficient binary search routine, which makes at most a logarithmic number of queries in the length of the interaction with the environment. Finally, the labeled states can be used to learn a classifier for predicting irreversible states, which can then be leveraged to proactively call for interventions. Our proposed framework PAINT can be used to adapt any value-based RL algorithm, in both episodic and non-episodic settings, to learn with minimal and proactive interventions. We compare PAINT to prior methods for autonomous RL and safe RL on a suite of continuous control tasks, and find that PAINT exhibits both better sample- and intervention-efficiency compared to existing methods. On challenging autonomous object manipulation tasks, PAINT only requires around 100 interventions while training for 3 million steps, which is up to $15\times$ fewer than those required by prior algorithms.

## 2 Related Work

Deployment of many RL algorithms in physical contexts is challenging, because they fail to avoid undesirable states in the environment and require human-provided resets between trials. Safe RL, reversibility-aware RL, and autonomous RL, which we review next, address parts of these problems.

**Safe RL.** The goal of our work is to learn to avoid irreversible states. Algorithms for safe RL also need to avoid regions of the state space, and achieve this by formulating a constrained optimization problem [10, 39, 37, 50] or by assigning low rewards to unsafe states [43, 41]. Another class of algorithms construct shielding-based policies that yield control to a backup policy if following the learning policy leads to an unsafe state [1, 3, 42, 40, 7, 43, 6]. Critically, however, all of these approaches assume that safety labels for each state can be queried freely at every time-step of training, whereas our objective is to minimize labeling requirements over training.

**Reversibility-aware RL.** Reversibility and reachability have been studied in the context of RL to avoid actions that lead to irreversible states [24, 23, 31, 15] or, conversely, to guide exploration towards difficult-to-reach states [33, 5]. Unlike prior work, our study of reversibility primarily focuses on the non-episodic setting to minimize the number of human interventions during learning. While prior methods are self-supervised, our experiments also find that our algorithm learns with significantly fewer interventions than prior methods by leveraging some binary reversibility labels.

**Autonomous RL.** Multiple prior works have also studied autonomy in RL, motivated by the fact that deployments of RL algorithms on real robots often require human-provided resets between episodes [13, 16, 14]. To avoid the supervision needed for episodic resets, prior work has proposed to learn controllers to return to specific state distributions, such as the initial state [19, 11], the uniform distribution over states [53] or demonstration states [36], adversarially learned distributions [47], or curriculum-based distributions [34]. However, most work in the reset-free setting assumes the agent's environment is reversible [30, 35, 34, 17], whereas we specifically tackle the setting where this is not the case. One notable exception is the Leave No Trace algorithm [11], which checks whether the agent has successfully returned back to the initial state distribution and requests an intervention otherwise. Our approach differs from Leave No Trace by requesting a reset based on the estimated reversibility of the state, which we find requires significantly fewer interventions in our evaluation.

**Human-in-the-loop learning.** Learning from human feedback has enabled RL agents to acquire complex skills that are difficult to encode in a reward function [22, 27, 45, 12, 4]. However, interactive RL algorithms are often to difficult to scale as they rely on feedback at every time-step of training. More feedback-efficient algorithms have learned reward models from human-provided preferences [2, 38, 46, 32, 8, 26, 44], which removes the need for constant feedback. Similarly, the interactive imitation learning learning has seen more query-efficient algorithms, which query expert actions based on the estimated risk or novelty of a visited state [51, 28, 21, 20]. While these algorithms augment the agent with human-provided preferences or expert actions, our approach leverages a different mode of feedback, that is, reversibility labels for visited states.

# 3   Reinforcement Learning in Irreversible Environments

Consider a Markov decision process $\mathcal{M} = (\mathcal{S}, \mathcal{A}, \mathcal{P}, r, \rho_0, \gamma)$ with state space $\mathcal{S}$, action space $\mathcal{A}$, transition dynamics $\mathcal{P} : \mathcal{S} \times \mathcal{A} \times \mathcal{S} \mapsto [0, 1]$, bounded reward function $r : \mathcal{S} \times \mathcal{A} \mapsto [R_{\min}, R_{\max}]$, initial state distribution $\rho_0 : \mathcal{S} \mapsto [0, 1]$ and discount factor $\gamma \in [0, 1)$. In this work, we build on the formalism of autonomous RL [35], but we remove the assumption that the environment is reversible, i.e., the MDP is no longer strongly connected (see example in [25, Chapter 38] and description below). The environment is initialized at $s_0 \sim \rho$ and an algorithm continually interacts with the environment till it requests the environment to be reset via an external intervention to state $s_0' \sim \rho$. Specifically, an algorithm $\mathbb{A} : \{s_i, a_i, s_{i+1}, r_i\}_{i=0}^{t-1} \mapsto (a_t, \pi_t)$ generates a sequence $(s_0, a_0, s_1, \dots)$ in $\mathcal{M}$, mapping the states, actions, and rewards seen till time $t-1$ to an action $a_t \in \mathcal{A} \cup \{a_{\texttt{reset}}\}$, and the current guess at the optimal policy $\pi_t : \mathcal{S} \times \mathcal{A} \mapsto [0, \infty)$. Here, $a_{\texttt{reset}}$ is a special action the agent can execute to reset the environment through extrinsic interventions, i.e. $\mathcal{P}(\cdot \mid s, a_{\texttt{reset}}) = \rho_0(\cdot)$.

A MDP is strongly connected if for all pairs of states $s_i, s_j \in \mathcal{S}$, there exists a policy $\pi$ such that $s_j$ has a non-zero probability of being visited when executing the policy $\pi$ from state $s_i$. This assumption can easily be violated in practice, for example, when a robot arm pushes an object out of its reach. At an abstract level, the agent has transitioned into a component of MDP that is not connected with the high reward states, and thus cannot continue making progress, as visualized in Figure 2. The agent can invoke an extrinsic agent (such as a human supervisor) through $a_{\texttt{reset}}$, and the extrinsic agent can reset the environment to a state from the initial state distribution. For example, the human supervisor can reset the object to the initial state, which is within the reach of the robot arm. For every state $s \in \mathcal{S}$, define $\mathcal{R}_\rho : \mathcal{S} \mapsto \{0, 1\}$ as the indicator whether the state $s$ is in the same component as the initial state distribution. State $s$ is defined to be reversible if $\mathcal{R}_\rho(s) = 1$, and irreversible if $\mathcal{R}_\rho(s) = 0$. We assume that the $\mathcal{R}_\rho$ is unknown, but can be queried for a state $s$.

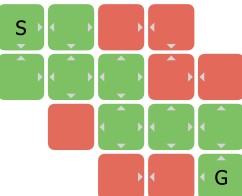

Figure 2: Example of an MDP with irreversible states (in red). The agent starts in the state 'S' and its goal is to reach the state 'G', which are connected.

While we do not assume that the MDP is strongly connected, we assume that the states visited by the optimal policy are in the same connected component as the initial state distribution. Under this assumption, we can design agents that can autonomously practice the task many times. Otherwise, the environment would need to be reset after every successful trial of the task.

The objective is to learn an optimal policy $\pi^* \in \arg\max_\pi J(\pi) = \arg\max_\pi \mathbb{E}[\sum_{t=0}^{\infty} \gamma^t r(s_t, a_t)]$. Note, $J(\pi)$ is approximated by computing the return when the policy is rolled out from $s_0 \sim \rho_0$. Algorithms are typically evaluated on the sample efficiency, that is minimizing

$\mathbb{D}(\mathbb{A}) = \sum_{t=0}^{\infty} J(\pi^*) - J(\pi_t)$. However, since we care about minimizing the human supervision required and resetting the environment can entail expensive human supervision, we will primarily evaluate algorithms on *intervention-efficiency*, defined as $\mathbb{I}(\mathbb{A}) = \sum_{k=0}^{\infty} J(\pi^*) - J(\pi_k)$, where $\pi_k$ is the policy learned after $k$ interventions.

## 4 Preliminaries

Episodic settings reset the environment to a state from the initial state distribution after every trial, typically after every few hundred steps of interaction with the environment. Such frequent resetting of the environment entails an extensive amount of external interventions, typically from a human. Prior works on autonomous RL have sought to reduce the supervision required for resetting the environments by learning a backward policy that resets the environment [11, 52, 34]. Meaningfully improving the autonomy of RL in irreversible environments requires us to curb the requirement of episodic resets first. While our proposed framework is compatible with any autonomous RL algorithm, we describe MEDAL [36], which will be used in our experiments.

MEDAL learns a forward policy $\pi_f$ and a backward policy $\pi_b$, alternately executed for a fixed number of steps in the environment. The forward policy maximizes the conventional cumulative task reward, that is $\mathbb{E}\left[\sum_{t=0}^{\infty} \gamma^t r(s_t, a_t)\right]$, and the backward policy minimizes the Jensen-Shannon divergence $\mathcal{D}_{\text{JS}}\left(\rho^b(s) \,\|\, \hat{\rho}^*(s)\right)$ between the marginal state distribution of the backward policy $\rho^b$ and the state distribution of the optimal forward policy $\hat{\rho}^*$, approximated by a small number of expert demonstrations. Thus, the backward policy keeps the agent close to the demonstration states, allowing the forward agent to try the task from a mix of easy and hard initial states. The proposed divergence can be minimized via the objective $\min_{\pi_b} \max_C \mathbb{E}_{s \sim \rho^*}\left[\log C(s)\right] + \mathbb{E}_{s \sim \rho^b}\left[\log(1 - C(s))\right]$, where $C : \mathcal{S} \mapsto [0, 1]$ is a classifier that maximizes the log-probability of states visited in the forward demonstrations, and minimizes the probability of the states visited by the backward policy. The optimization problem for the backward policy can be written as a RL problem:

$$\min_{\pi_b} \mathbb{E}_{s \sim \rho^b}\left[\log\left(1 - C(s)\right)\right] = \max_{\pi_b} \mathbb{E}\left[-\sum_{t=0}^{\infty} \gamma^t \log\left(1 - C(s)\right)\right] \tag{1}$$

where $\pi_b$ maximizes the reward function $r(s, a) = -\log\left(1 - C(s)\right)$. Correspondingly, $C(s)$ is trained to discriminate between states visited by the backward policy and the demonstrations.

## 5 Proactive Agent Interventions for Autonomous Reinforcement Learning

To minimize human monitoring and supervision when an agent is learning in an environment with irreversible states, the agent needs to (a) learn to avoid irreversible states over the course of training and (b) learn to detect and *request* an intervention whenever the agent is stuck. For the former, we first describe a simple modification to the reward function to explicitly penalize visitation of irreversible states in Section 5.1. However, such a modification requires the knowledge of reversibility of the visited states, which is not known apriori. We learn a classifier to estimate reversibility, proposing a label-efficient algorithm to query reversibility labels of visited states in Section 5.2. Since both the dynamics and the set of irreversible states are unknown apriori, the agent will inevitably still visit irreversible states as a part of the exploration. To ensure that a human does not have to monitor the agent throughout training, the agent should have a mechanism to decide and request for an intervention. We discuss such a mechanism in Section 5.3. Finally, we put together all these components in Section 5.4 for our proposed framework **P**roactive **A**gent **INT**erventions (PAINT), an overview of which is given in Figure 3.

### 5.1 Penalizing Visitation of Irreversible States

Our goal is to penalize visitation of irreversible states by ensuring all actions leading to irreversible states are 'worse' than those leading to reversible states. To this end, we adapt the reward-penalty framework from safe RL [41] for learning in the presence of irreversible states. For a transition $(s, a, s')$, consider a surrogate reward function $\tilde{r}$:

$$\tilde{r}(s, a) = \begin{cases} r(s, a), & \mathcal{R}_\rho(s') = 1 \\ R_{\min} - \epsilon, & \mathcal{R}_\rho(s') = 0 \end{cases} \tag{2}$$

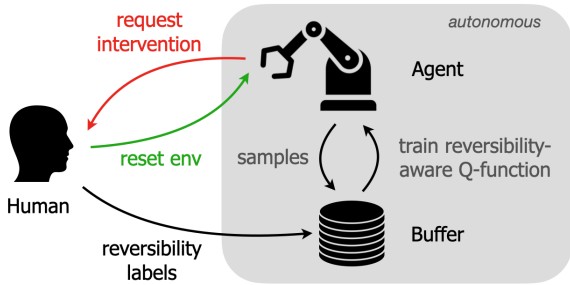

Figure 3: Overview of our framework PAINT for minimizing human monitoring and supervision when learning in the presence of irreversible states. The agent proactively requests interventions, freeing the human from active monitoring of training. When an intervention is requested, the human resets the environment and provides reversibility labels for the latest experience since the previous intervention.

**Algorithm 1:** Reversibility Labeling via Binary Search

**input:** $\tau = \{s_i\}_{i=0}^{T}$; // unlabeled trajectory
**while** $len(\tau) > 0$ **do**
  $m \leftarrow \lfloor len(\tau)/2 \rfloor$; // get midpoint
  // query for midpoint
  **if** $\mathcal{R}_\rho(s_m) = 1$ **then**
    // label first half reversible and query for second half
    label $\{s_i\}_{i=0}^{m}$ as 1;
    $\tau \leftarrow \{s_i\}_{i=m+1}^{len(\tau)}$;
  **else**
    // label second half irreversible and query for first half
    label $\{s_i\}_{i=m+1}^{len(\tau)}$ as 0;
    $\tau \leftarrow \{s_i\}_{i=0}^{m}$;

Whenever the next state $s'$ is a reversible state, the agent gets the environment reward. Otherwise if it has entered an irreversible, it gets a constant reward $R_{\min} - \epsilon$ that is worse than any reward given out by the environment. Whenever an agent enters an irreversible state, it will continue to remain in an irreversible state and get a constant reward of $R_{\min} - \epsilon$. Therefore, the $Q$-value whenever $\mathcal{R}_\rho(s') = 0$ is given by:

$$Q^\pi(s,a) = \mathbb{E}\left[ \sum_{t=0}^{\infty} \gamma^t \tilde{r}(s_t, a_t) \mid s_0 = s, a_0 = a \right] = (R_{\min} - \epsilon) \sum_{t=0}^{\infty} \gamma^t = \frac{R_{\min} - \epsilon}{1 - \gamma}$$

This observation allows us to bypass the need to perform Bellman backups on irreversible states, and instead directly regress to the $Q$-value. More specifically, we can rewrite the loss function for the $Q$-value function as, $\ell(Q) = \mathbb{E}_{(s,a,s',r)\sim\mathcal{D}}[Q(s,a) - \mathcal{B}^\pi Q(s,a)]$, where the application of Bellman backup operator $\mathcal{B}^\pi Q(s,a)$ can be expanded as:

$$\mathcal{B}^\pi Q(s,a) = \begin{cases} r(s,a) + \gamma \mathbb{E}_{a'\sim\pi(\cdot|s')}\hat{Q}(s',a'), & \mathcal{R}_\rho(s') = 1 \\ (R_{\min} - \epsilon)/(1 - \gamma), & \mathcal{R}_\rho(s') = 0 \end{cases} \tag{3}$$

$$= \mathcal{R}_\rho(s')\left( r(s,a) + \gamma \mathbb{E}_{a'\sim\pi(\cdot|s')}\hat{Q}(s',a') \right) + (1 - \mathcal{R}_\rho(s'))\frac{R_{\min} - \epsilon}{1 - \gamma} \tag{4}$$

Here, $\mathcal{D}$ denotes the replay buffer, $\hat{Q}$ denotes the use of target networks commonly used in $Q$-learning algorithms to stabilize training when using neural networks as function approximators [29]. This surrogate reward function and the modified Bellman operator can be used for any value-based RL algorithm in both episodic and autonomous RL settings. The hyperparameter $\epsilon$ controls how aggressively the agent is penalized for visiting irreversible states. In general, $Q$-values for actions leading to irreversible states will be lower than those keeping the agent amongst reversible states, encouraging the policies to visit irreversible states fewer times over the course of training. More details and proofs can be found in Appendix A.1.1.

### 5.2 Estimating Reversibility

In general, $\mathcal{R}_\rho$ is not known apriori and will have to be estimated. We define $\hat{\mathcal{R}}_\rho : \mathcal{S} \mapsto [0,1]$ as an estimator of the reversibility of a state $s \in \mathcal{S}$. We can then define an empirical Bellman backup operator $\hat{\mathcal{B}}^\pi$ from equation 4 by replacing $\mathcal{R}_\rho$ with the estimator $\hat{\mathcal{R}}_\rho$:

$$\hat{\mathcal{B}}^\pi Q(s,a) = \mathbb{E}_{s'\sim\mathcal{P}(\cdot|s,a)}\left[ \hat{\mathcal{R}}_\rho(s')\left( r(s,a) + \gamma \mathbb{E}_{a'\sim\pi(\cdot|s')}Q(s',a') \right) + \left(1 - \hat{\mathcal{R}}_\rho(s')\right)\frac{R_{\min} - \epsilon}{1 - \gamma} \right]$$

Analogous to $\hat{\mathcal{B}}^\pi$, we can define the empirical Bellman optimality operator $\hat{\mathcal{B}}^*$ (Eq 12) for value iteration when using $\hat{\mathcal{R}}_\rho$. The following theorem bounds the suboptimality of the policy learned by value iteration under $\hat{\mathcal{B}}^*$:

**Theorem 5.1.** *Let $\pi^*$ denote the optimal policy and $Q^*$ denote the corresponding optimal Q-value function. Let $\hat{\pi}^*$ denote the optimal policy returned by empirical Bellman optimality operator $\hat{\mathcal{B}}^*$. Assuming $\|\mathcal{R}_\rho - \hat{\mathcal{R}}_\rho\|_\infty \leq \delta$,*

$$Q^{\hat{\pi}^*}(s,a) \geq Q^*(s,a) - \frac{2\delta\left(R_{max} - R_{min} + \epsilon\right)}{(1-\gamma)^2}$$

*for all $(s,a) \in \mathcal{S} \times \mathcal{A}$.*

The proof and related discussion can be found in Appendix A.1.2. The result guarantees that closer $\hat{\mathcal{R}}_\rho$ is to $\mathcal{R}_\rho$ under the $\infty$-norm, the closer $\hat{\pi}^*$ is to $\pi^*$. To this end, we propose to learn $\hat{\mathcal{R}}_\rho$ by minimizing the binary cross-entropy loss $\ell(\hat{\mathcal{R}}_\rho) = -\mathbb{E}_{s \sim \mathcal{D}}\left[\mathcal{R}_\rho(s) \log \hat{\mathcal{R}}_\rho(s) + (1 - \mathcal{R}_\rho(s)) \log(1 - \hat{\mathcal{R}}_\rho(s))\right]$, where the states $s \sim \mathcal{D}$ represent the states visited by the agent.

Minimizing $\ell(\hat{\mathcal{R}}_\rho)$ requires the reversibility labels $\mathcal{R}_\rho(s)$ for $s \sim \mathcal{D}$. Since labeling requires supervision, it is critical to query $\mathcal{R}_\rho$ efficiently. Given a trajectory of states $\tau = (s_0, s_1, \ldots s_T)$, a naïve approach would be to query the labels $\mathcal{R}_\rho(s_i)$ for all states $s_i$, leading to $\mathcal{O}(T)$ queries per trajectory. However, observe that we have the following properties: (a) all states *following* an irreversible state will be irreversible and (b) all states *preceding* a reversible state will be reversible. It follows from these properties that every trajectory can be split into a reversible segment $\tau_r = (s_0, s_1, \ldots s_k)$ and an irreversible segment $\tau_{\sim r} = (s_{k+1}, \ldots s_T)$, where the irreversible segment $\tau_{\sim r}$ can be empty potentially. Identifying $s_{k+1}$, the first irreversible state, generates the labels for the entire trajectory automatically. Fortunately, we can construct a scheme based on binary search to identify $s_{k+1}$ in $\mathcal{O}(\log T)$ queries: $s_{k+1}$ occurs after the midpoint of the trajectory if the midpoint is reversible, otherwise it occurs before it. The pseudocode for this routine is given in Alg 1.

The total number of labels required would be $\mathcal{O}\left(N \log |\tau|_{max}\right)$, where $N$ is the number of trajectories in the replay buffer $\mathcal{D}$ and the $|\tau|_{max}$ denotes the maximum length of the trajectory. This represents a reduction in label requirement of $\mathcal{O}\left(N |\tau|_{max}\right)$ by prior safe RL methods. Furthermore, agent trains to avoid irreversible states, resulting in fewer and longer trajectories over the course of training. Thus, labeling reduces over time because the labels required is linear in $N$ and logarithmic in $|\tau|_{max}$.

### 5.3 Proactive Interventions

Despite trying to avoid irreversible states via reward penalties, an agent will inevitably encounter some irreversible states due to exploratory behaviors. It is critical that the agent proactively asks for help in such situations, so that a human does not need to constantly monitor the training process. More specifically, an agent should request an intervention when it is an irreversible state. Since $\mathcal{R}_\rho(s)$ is not available, the agent again needs to estimate the reversibility of the state. It is natural to reuse the learned reversibility estimator $\hat{\mathcal{R}}_\rho$ for this purpose. We propose the following rule: the agent executes $a_{\texttt{reset}}$ whenever the reversibility classifier's prediction falls below 0.5, i.e., $\hat{\mathcal{R}}_\rho(s) < 0.5$.

### 5.4 Putting it Together

With the key components in place, we summarize our proposed framework. High-level pseudocode is given in Alg. 2, and a more detailed pseudocode is deferred to Appendix A.2.

PAINT can modify any value-based RL algorithm, in both episodic and autonomous settings. This description and Alg. 2 focus on the latter setting, although adapting it to the episodic setting is straightforward. The agent's interaction with the environment consists of a sequence of trials that end whenever the environment is reset to a state $s \sim \rho_0$. During each trial, the agent operates autonomously, and the Bellman update for the critic is modified according to the empirical Bellman backup $\hat{\mathcal{B}}^\pi$. Whenever the

---

**Algorithm 2:** PAINT

**input:** $\mathbb{P}$; // agent, params abstracted away
initialize $\hat{\mathcal{R}}_\rho, \mathcal{D}$; // rev classifier, replay buffer
**while** *not done* **do**
    $s \sim \rho_0$; // reset environment
    // continue till classifier detects irreversibility
    **while** $\hat{\mathcal{R}}_\rho(s) > 0.5$ **do**
        // step in the environment
        $a \sim \mathbb{P}(s), s \sim \mathcal{P}(\cdot \mid s, a)$;
        // update replay buffer and agent
        update $\mathcal{D}, \mathbb{P}$;
    // optionally explore environment
    **for** *explore steps* **do**
        $a \sim \text{unif}(\mathcal{A}), s \sim \mathcal{P}(\cdot \mid s, a)$;
        update $\mathcal{D}$;
    // reversibility labels via binary search
    update reversibility labels in $\mathcal{D}$;
    // train classifier on all labeled data, new and old
    train $\hat{\mathcal{R}}_\rho$;

---

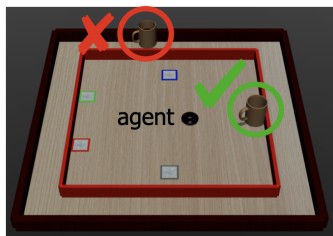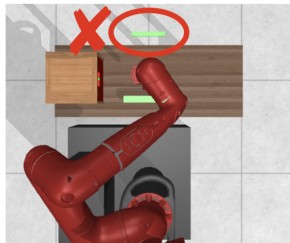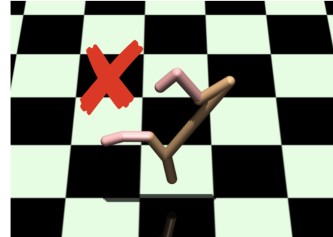

Figure 4: A subset of our evaluation tasks: Tabletop Manipulation, Peg Insertion, and Half-Cheetah Velocity. Irreversible states in the first two environments are when the agent drops the object outside the red boundary (*left*) and off of the table (*middle*). The cheetah is in an irreversible state whenever it is flipped over (*right*).

reversibility classifier $\hat{\mathcal{R}}_\rho < 0.5$, parameterized
as a neural network, the agent requests an intervention. The agent can execute a fixed number of exploration steps after requesting an intervention and before the intervention is performed. Whenever the classifier predicts an irreversible state correctly, these exploration steps can help the agent gather more information about irreversible states. At the time of the intervention, all new states visited since the previous intervention are labeled for reversibility via Algorithm 1. Finally, the reversibility classifier is trained on all the labeled data before the environment is reset to a state $s \sim \rho_0$ for the next trial. Full implementation details can be found in Appendix A.4.

The agent is provided reversibility labels only when the external reset is provided. This simplifies supervision as the human can reset the environment and provide labels at the same time. This means the replay buffer $\mathcal{D}$ will contain states with and without reversibility labels, since states from the current trial will not yet have labels. We use Eq. 4 for states that have reversibility labels to avoid errors from the classifier affecting the critic update and use $\hat{\mathcal{B}}^\pi$ for those that do not have labels.

## 6 Experiments

We design several experiments to study the efficiency of our algorithm in terms of the required number of reset interventions and number of queried reversibility labels. Our code and videos of our results are at: https://sites.google.com/view/proactive-interventions.

### 6.1 Experimental Setup

**Environments.** To illustrate the wide applicability of our method, we design environments that represent three distinct RL setups: episodic, forward-backward, and continuing.

- **Maze** (episodic). A 2D continuous maze environment with trenches, which represent groups of connected irreversible states. The agent can fall into a trench, and once entered, it can roam freely within the trench but cannot leave it without an environment reset. In this task, resets are infrequently provided to the agent after 500 time-steps.
- **Tabletop Organization** [35] (forward-backward). The agent must grasp the mug and put it down on one of the four goal positions. Dropping the mug outside of the red boundary is irreversible.
- **Peg Insertion** [35] (forward-backward). The agent must insert the peg into the goal but can potentially drop it off the table, which is irreversible.
- **Half-Cheetah Vel** [9] (continuing). The agent must run at the specified target velocity, which changes every 500 steps, and can potentially flip over onto its back, which is irreversible.

We visualize and fully describe each environment in Fig. 4 and in Appendix A.3 respectively.

**Comparisons.** In the episodic and continuing settings, we consider safe RL baselines that rely on reversibility labels at every time-step of training.

- **Safe Model-Based Policy Optimization (SMBPO)** [41]. This comparison implements the modified Bellman operator defined in Eqn. 4 in Section 5.1, using the true reversibility labels.
- **Safety Q-functions for RL (SQRL)** [37]. A safe RL method that trains a safety critic, which estimates the future probability of entering an irreversible state for a safety-constrained policy.

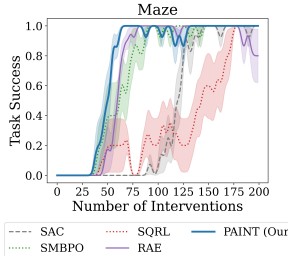

Figure 5: (*left*) Task success versus interventions. Shaded regions denote the standard error over 5 seeds. (*right*) Predictions generated by our reversibility classifier, where the purple region is predicted to be reversible.

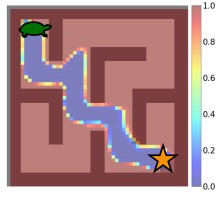

| Task | Method | Labels |
|---|---|---|
| Maze | SMBPO/SQRL | 200K |
| | PAINT (Ours) | $3260 \pm 12$ |
| Tabletop | PAINT (Ours) | $1021 \pm 69$ |
| Peg Insertion | PAINT (Ours) | $2083 \pm 149$ |
| Cheetah | SMBPO w Term. | 3M |
| | PAINT (Ours) | $8748 \pm 3762$ |

Figure 6: Number of queried reversibility labels. For our method, we average the number of labels used across 5 seeds and report the standard error.

In the forward-backward setting, we consider methods designed for the autonomous learning setup. These methods do not require any reversibility labels. Hence, our goal here is to compare the reset-efficiency of our method to prior work.

- **Leave No Trace (LNT)** [11]. An autonomous RL method that jointly trains a forward policy and reset policy. When the reset policy fails to return to the initial state, the agent requests a reset.
- **Matching Expert Distributions for Autonomous Learning (MEDAL)** [36]. This method trains a reset policy that returns to the distribution of demonstration states provided for the forward policy. MEDAL does not have a built-in intervention rule.

In all tasks, we compare to a recently proposed reversibility-aware RL method, **Reversibility-Aware Exploration (RAE)** [15], which does not require any reversibility labels. It instead trains a self-supervised reversibility estimator to predict whether a state transition $(s, \tilde{s})$ is more likely than the reverse $(\tilde{s}, s)$. We augment RAE with an intervention rule, similar to our method, defined in terms of predictions from its self-supervised classifier. In the forward-backward setting, we train both the forward and backward policies with RAE. Finally, we also evaluate **Episodic RL**, which represents the typical RL setup with frequent resets and thus provides an upper-bound on task success. In Appendix A.4, we provide full implementation details of each comparison, and in Appendix A.5, we discuss the set of assumptions made by each comparison.

## 6.2 Main Results

In Fig. 5 (*left*) and Fig. 7, we plot the task success versus the number of interventions in the 4 tasks. For methods that use reversibility labels, we report the total number of labels queried in Table 6.

**Maze.** While the safe RL methods, SMBPO and SQRL, require reversibility labels at every time-step, our approach PAINT only requires on average 3260 queries to label all 200K states visited. In Fig. 5 (right), we visualize predictions from our reversibility classifier at the end of training, where zero predicts 'reversible' and one predicts 'irreversible'. The classifier correctly identifies the path that leads to the goal as reversible. Interestingly, it classifies all other regions as stuck states, including the states that *are* reversible. Because these states are irrelevant to the task, however, classifying them as irreversible, and therefore to be avoided, is advantageous to our policy as it reduces its area of exploration.

**More complex domains.** In the Tabletop Organization and Peg Insertion tasks, each agent is reset every 200K and 100K time-steps, per the EARL benchmark [35]. However, we allow agents to request earlier resets, and under this setting, we compare PAINT to other methods that implement intervention rules. Compared to Leave No Trace and Reversibility-Aware Exploration, PAINT requires significantly fewer resets—80 and 124 resets respectively, which corresponds to roughly **one intervention for every 25K steps**. Importantly, the number of interventions plateaus as training progresses, and the agent requires fewer and fewer resets over time (see Appendix A.6 for additional plots of number of interventions versus time-steps). The exception is MEDAL (green segment near the origin), which is not equipped with an early termination rule and so only uses 10 interventions total. However, it also fails to make meaningful progress on the task with few resets.

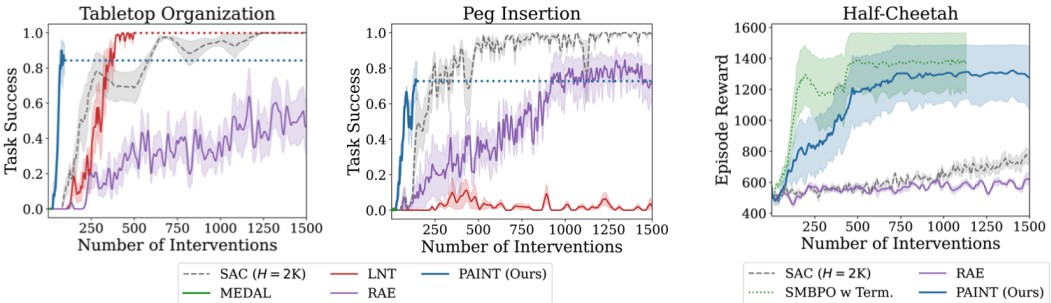

Figure 7: Task success versus interventions averaged over 5 seeds. Methods with stronger assumptions, i.e., SAC resets every $H$ steps and SMBPO requires labels at every time-step, are dotted. Note the short green segment near the origin representing MEDAL.

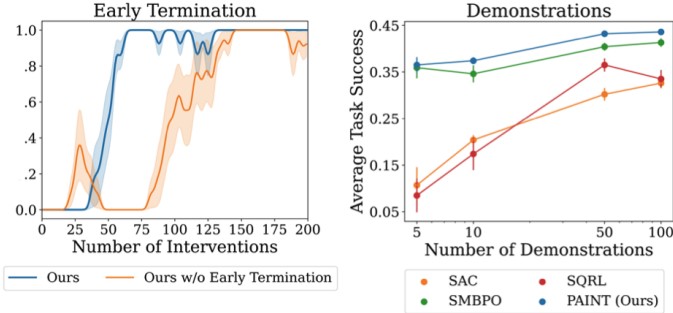

Figure 8: (*left*) After removing the early termination condition, which initiates random exploration, we find that PAINT learns less efficiently. (*right*) Varying the number of demonstrations suggests that PAINT and SMBPO are robust to the amount of available demonstrations.

On the continuing Half-Cheetah task, agents do not receive any resets, unless specifically requested. Here, we compare PAINT to SMPBO with early termination, an **oracle** version of our method, which assumes that reversibility labels are available at *every* time-step and immediately requests an intervention if the agent is flipped over. PAINT converges to its final performance after around 750 resets, on par with the number of resets required by SMPBO with early termination. On the other hand, a standard episodic RL agent, which receives resets at every 2K steps, and RAE, which trains a self-supervised classifier, learn significantly slower with respect to the number of interventions.

### 6.3 Ablations and Sensitivity Analysis

**Early termination**. In the episodic Maze setting, our algorithm switches to a uniform-random policy for the remainder of the episode if the termination condition is met. In Fig. 8 (*left*), we plot the performance without early termination, i.e., running the agent policy for the full episode. Taking random explorations, after the agent believes it has entered an irreversible state, significantly helps our method, as it increases the number and diversity of irreversible states the agent has seen.

**Varying the number of demonstrations**. Our method leverages demonstrations in a subset of environments. While we provide these demonstrations to all comparisons as well, we want to study how much our method relies on them. We plot the average task success during training versus number of demonstrations in Fig. 8 (*right*). While PAINT and SMBPO are robust to the amount, alternative methods tend to achieve significantly lower success when given fewer demonstrations.

## 7 Discussion

In this work, we sought to build greater autonomy into RL agents, particularly in irreversible environments. We proposed an algorithm, PAINT, that learns to detect and avoid irreversible states, and proactively requests an intervention when in an irreversible state. PAINT leverages reversibility

labels to learn to identify irreversible states more quickly, and improves upon existing methods on a range of learning setups in terms of task success, reset-efficiency, and label-efficiency.

Despite these improvements, PAINT has multiple important limitations. In environments where irreversible states are not encountered until further into training, the reversibility classifier may produce false positives which would significantly delay the next intervention. Further, while PAINT is far more label-efficient than prior safe RL methods, it still requires around thousands of reversibility labels. We expect that this limitation may be mitigated with more sophisticated querying strategies, e.g. that take into account the classifier's confidence. Finally, we hope that future work can validate the ability for reversibility aware techniques to improve the autonomy of real robotic learning systems.

## Acknowledgments and Disclosure of Funding

AX was supported by an NSF Graduate Research Fellowship. The work was also supported by funding from Google, Schmidt Futures, and ONR grants N00014-21-1-2685 and N00014-20-1-2675. The authors would also like to thank members of the IRIS Lab for helpful feedback on an early version of this paper.

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
