# A  Appendix

## A.1  Proofs

### A.1.1  Penalizing Visitation of Irreversible States

The high level goal of this section is to show that $Q$-values prefer the actions leading to reversible states when using the surrogate reward function (Eq 2). We first present the result for deterministic dynamics, and then give a more restricted proof for stochastic dynamics. For deterministic dynamics, define $\mathcal{S}_{\text{rev}} = \{(s, a) \mid \mathcal{R}_\rho(s') = 1\}$.

**Theorem A.1.** *Let $(s, a) \in \mathcal{S}_{rev}$ denote a state-action leading to a reversible state and let $(s_-, a_-) \notin \mathcal{S}_{rev}$ denote a state-action pair leading to an irreversible state. Then, for all such pairs*

$$Q^\pi(s, a) > Q^\pi(s_-, a_-)$$

*for all $\epsilon > 0$ and for all policies $\pi$.*

*Proof.* By definition, the reward function is bounded, i.e., $r(s', a') \in [R_{\min}, R_{\max}]$ for any $(s', a') \in \mathcal{S} \times \mathcal{A}$. For any $(s', a') \in \mathcal{S}_{\text{rev}}$, we have the following (using equation 2):

$$\tilde{r}(s', a') = r(s', a') \geq R_{\min} > R_{\min} - \epsilon$$

and for any $(s', a') \notin \mathcal{S}_{\text{rev}}$, we have:

$$\tilde{r}(s', a') = R_{\min} - \epsilon$$

This simplifies to $\tilde{r}(s', a') \geq R_{\min} - \epsilon$ for any $(s', a') \in \mathcal{S} \times \mathcal{A}$. From the definition of $Q^\pi$, we have

$$Q^\pi(s', a') = \mathbb{E}\Big[\sum_{t=0}^{\infty} \gamma^t \tilde{r}(s_t, a_t) \big| s_0 = s', a_0 = a'\Big] \geq \sum_{t=0}^{\infty} \gamma^t (R_{\min} - \epsilon) = \frac{R_{\min} - \epsilon}{1 - \gamma} \tag{5}$$

where the lower bound of $Q^\pi(s', a')$ is achieved if $(s', a') \notin \mathcal{S}_{\text{rev}}$. For $(s, a) \in \mathcal{S}_{\text{rev}}$,

$$Q^\pi(s, a) = \tilde{r}(s, a) + \gamma \mathbb{E}_{a' \sim \pi(\cdot | s')} \left[ Q^\pi(s', a') \right]$$

$$\geq \tilde{r}(s, a) + \gamma \frac{R_{\min} - \epsilon}{1 - \gamma}$$

$$\geq R_{\min} + \gamma \frac{R_{\min} - \epsilon}{1 - \gamma} \qquad \left[ \tilde{r}(s, a) = r(s, a) \geq R_{\min}, \text{ since } (s, a) \in \mathcal{S}_{\text{rev}} \right]$$

$$= \epsilon + (R_{\min} - \epsilon) + \gamma \frac{R_{\min} - \epsilon}{1 - \gamma} = \epsilon + \frac{R_{\min} - \epsilon}{1 - \gamma} = \epsilon + Q^\pi(s_-, a_-)$$

where $(s_-, a_-) \notin \mathcal{S}_{\text{rev}}$, implying $Q^\pi(s_-, a_-) = \frac{R_{\min} - \epsilon}{1 - \gamma}$. This concludes the proof as $Q^\pi(s, a) > Q^\pi(s_-, a_-)$ for $\epsilon > 0$. $\qquad \square$

To extend the discussion to stochastic dynamics, we redefine the reversibility set as $\mathcal{S}_{\text{rev}} = \{(s, a) \mid \mathbb{P}\left( \mathcal{R}_\rho(s') = 1 \right) \geq \eta_1 \}$, i.e, state-action pairs leading to a reversible state with at least $\eta_1$ probability and let $\mathcal{S}_{\text{irrev}} = \{(s, a) \mid \mathbb{P}\left( \mathcal{R}_\rho(s') = 1 \right) \leq \eta_2 \}$ denote the set of state-action pairs leading to irreversible states with at most $\eta_2$ probability. The goal is to show that actions leading to reversible states with high probability will have higher $Q$-values than those leading to irreversible states with high probability. The surrogate reward function $\tilde{r}$ is defined as:

$$\tilde{r}(s, a, s') = \begin{cases} r(s, a, s'), & \mathcal{R}_\rho(s') = 1 \\ R_{\min} - \epsilon, & \mathcal{R}_\rho(s') = 0 \end{cases} \tag{6}$$

**Theorem A.2.** *Let $(s, a) \in \mathcal{S}_{rev}$ denote a state-action leading to a reversible state with at least $\eta_1$ probability and let $(s_-, a_-) \in \mathcal{S}_{irrev}$ denote a state-action pair leading to an irreversible state with at least $(1 - \eta_2)$ probability. Assuming $\eta_1 > \eta_2/(1 - \gamma)$,*

$$Q^\pi(s, a) > Q^\pi(s_-, a_-)$$

*for all $\epsilon > \frac{\eta_2}{\eta_1 - \gamma \eta_1 - \eta_2} \left( R_{max} - R_{min} \right)$ and for all policies $\pi$.*

*Proof.* For any $(s, a) \in \mathcal{S}_{\text{rev}}$, we have the following (using equation 6):

$$Q^\pi(s, a) = \mathbb{P}\left(\mathcal{R}_\rho(s') = 1\right)\left(r(s, a) + \gamma \mathbb{E}_{a' \sim \pi(\cdot|s')}\left[Q^\pi(s', a')\right]\right) + \mathbb{P}\left(\mathcal{R}_\rho(s') = 0\right)\frac{R_{\min} - \epsilon}{1 - \gamma}$$

$$\geq \mathbb{P}\left(\mathcal{R}_\rho(s') = 1\right)\left(R_{\min} + \gamma\frac{R_{\min} - \epsilon}{1 - \gamma}\right) + \mathbb{P}\left(\mathcal{R}_\rho(s') = 0\right)\frac{R_{\min} - \epsilon}{1 - \gamma}$$

$$\geq \eta_1\left(R_{\min} + \gamma\frac{R_{\min} - \epsilon}{1 - \gamma}\right) + (1 - \eta_1)\frac{R_{\min} - \epsilon}{1 - \gamma}$$

$$= \eta_1 R_{\min} + (1 + \gamma\eta_1 - \eta_1)\frac{R_{\min} - \epsilon}{1 - \gamma} \tag{7}$$

where $\mathbb{P}(\mathcal{R}_\rho(s') = 1) \geq \eta_1$. For any $(s, a) \in \mathcal{S}_{\text{irrev}}$,

$$Q^\pi(s, a) = \mathbb{P}\left(\mathcal{R}_\rho(s') = 1\right)\left(r(s, a) + \gamma \mathbb{E}_{a' \sim \pi(\cdot|s')}\left[Q^\pi(s', a')\right]\right) + \mathbb{P}\left(\mathcal{R}_\rho(s') = 0\right)\frac{R_{\min} - \epsilon}{1 - \gamma}$$

$$\leq \mathbb{P}\left(\mathcal{R}_\rho(s') = 1\right)\frac{R_{\max}}{1 - \gamma} + \mathbb{P}\left(\mathcal{R}_\rho(s') = 0\right)\frac{R_{\min} - \epsilon}{1 - \gamma}$$

$$\leq \eta_2\frac{R_{\max}}{1 - \gamma} + (1 - \eta_2)\frac{R_{\min} - \epsilon}{1 - \gamma} \tag{8}$$

as $\mathbb{P}(\mathcal{R}_\rho(s') = 1) \leq \eta_2$) by definition of $\mathcal{S}_{\text{irrev}}$. Under the assumption that $\eta_1 > \eta_2/(1-\gamma)$, whenever

$$\eta_1 R_{\min} + (1 + \gamma\eta_1 - \eta_1)\frac{R_{\min} - \epsilon}{1 - \gamma} > \eta_2\frac{R_{\max}}{1 - \gamma} + (1 - \eta_2)\frac{R_{\min} - \epsilon}{1 - \gamma}$$

$$\eta_1 R_{\min} - \eta_2\frac{R_{\max}}{1 - \gamma} > (\eta_1 - \gamma\eta_1 - \eta_2)\frac{R_{\min} - \epsilon}{1 - \gamma}$$

$$\eta_1(1 - \gamma)R_{\min} - \eta_2 R_{\max} > (\eta_1 - \gamma\eta_1 - \eta_2)(R_{\min} - \epsilon)$$

$$\epsilon > \frac{\eta_2}{(\eta_1 - \gamma\eta_1 - \eta_2)}\left(R_{\max} - R_{\min}\right),$$

ensures that $Q^\pi(s, a) > Q^\pi(s_-, a_-)$ for all $(s, a) \in \mathcal{S}_{\text{rev}}$ and for all $(s_-, a_-) \in \mathcal{S}_{\text{irrev}}$, finishing the proof. $\qquad\square$

The above proof guarantees that for actions that lead to reversible states with high probability will have higher $Q$-values than actions leading to irreversible states with high probabilities for *all policies*, even under stochastic dynamics. The restrictive assumption of $\eta_1 > \eta_2/(1 - \gamma)$ is required to ensure that the $Q$-value for worst $(s, a) \in \mathcal{S}_{\text{rev}}$ is better than the best $(s, a) \in \mathcal{S}_{\text{irrev}}$. An alternate analysis can be found in [41, Appendix A.2], where the guarantees are only given for the optimal $Q$-functions but under less stringent assumptions. Improved guarantees are deferred to future work.

### A.1.2 On Empirical Bellman Backup Operator

The empirical Bellman backup operator was introduced in subsection 5.2. In this section, we prove that it is a contraction, and analyze the convergence under empirical Bellman backup. For $\hat{\mathcal{R}}_\rho : \mathcal{S} \mapsto [0, 1]$, the empirical Bellman backup operator can be written as:

$$\hat{\mathcal{B}}^\pi Q(s, a) = \mathbb{E}_{s' \sim \mathcal{P}(\cdot|s,a)}\left[\hat{\mathcal{R}}_\rho(s')\left(r(s, a) + \gamma\mathbb{E}_{a' \sim \pi(\cdot|s')}Q(s', a')\right) + \left(1 - \hat{\mathcal{R}}_\rho(s')\right)\frac{R_{\min} - \epsilon}{1 - \gamma}\right] \tag{9}$$

**Theorem A.3.** *Empirical Bellman backup operator in equation 9 is a contraction under the $L^\infty$ norm.*

*Proof.* For all $(s, a) \in \mathcal{S} \times \mathcal{A}$, we have the following:

$$\left|\hat{\mathcal{B}}^\pi Q(s, a) - \hat{\mathcal{B}}^\pi Q'(s, a)\right| = \gamma \left|\mathbb{E}_{s' \sim \mathcal{P}(\cdot|s, a)} \left[\hat{\mathcal{R}}_\rho(s') \mathbb{E}_{a' \sim \pi(\cdot|s')} \left[Q(s', a') - Q'(s', a')\right]\right]\right|$$

$$\leq \gamma \left|\mathbb{E}_{s' \sim \mathcal{P}(\cdot|s, a)} \left[\mathbb{E}_{a' \sim \pi(\cdot|s')} \left[Q(s', a') - Q'(s', a')\right]\right]\right|$$

$$\left(\text{since } \hat{\mathcal{R}}_\rho(s') \in [0, 1]\right)$$

$$\leq \gamma \max_{(s'', a'') \in \mathcal{S} \times \mathcal{A}} |Q(s'', a'') - Q'(s'', a'')|$$

$$= \gamma \|Q - Q'\|_\infty$$

Since this holds for all $(s, a) \in \mathcal{S} \times \mathcal{A}$, this implies:

$$\|\hat{\mathcal{B}}^\pi Q - \hat{\mathcal{B}}^\pi Q'\|_\infty = \max_{(s, a) \in \mathcal{S} \times \mathcal{A}} |\hat{\mathcal{B}}^\pi Q(s, a) - \hat{\mathcal{B}}^\pi Q'(s, a)| \leq \gamma \|Q - Q'\|_\infty$$

The discount factor $\gamma < 1$, proving our claim. □

For a policy $\pi$, let $Q^\pi$ be the true $Q$-value function computed using conventional Bellman backup $\mathcal{B}^\pi$ and $\hat{Q}^\pi$ be the $Q$-values computed using the empirical Bellman backup $\hat{\mathcal{B}}^\pi$. Being fixed point of the operators, we have $Q^\pi = \mathcal{B}^\pi Q^\pi$ and $\hat{Q}^\pi = \hat{\mathcal{B}}^\pi \hat{Q}^\pi$. The following theorem relates the two:

**Lemma A.4.** *Assuming that $\|\mathcal{R}_\rho - \hat{\mathcal{R}}_\rho\|_\infty \leq \delta$, the difference between true $Q$-values and empirical $Q$-values for any policy $\pi$ obeys the following inequality:*

$$\left|Q^\pi(s, a) - \hat{Q}^\pi(s, a)\right| \leq \frac{\delta \left(R_{max} - R_{min} + \epsilon\right)}{(1 - \gamma)^2}$$

*Proof.* Since $Q^\pi$ is the fixed point of $\mathcal{B}^\pi$ and $\hat{Q}^\pi$ is the fixed point of $\hat{\mathcal{B}}^\pi$, we can write:

$$\left|Q^\pi(s, a) - \hat{Q}^\pi(s, a)\right| = \left|\mathcal{B}^\pi Q^\pi - \hat{\mathcal{B}}^\pi \hat{Q}^\pi\right|$$

$$= \left|\mathbb{E}_{s' \sim \mathcal{P}(\cdot|s, a)} \left[\left(\mathcal{R}_\rho(s') - \hat{\mathcal{R}}_\rho(s')\right) \left(r(s, a) - \frac{R_{min} - \epsilon}{1 - \gamma}\right)\right.\right.$$

$$\left.\left. + \gamma \mathbb{E}_{a' \sim \pi(\cdot|s)} \left[\mathcal{R}_\rho(s') Q^\pi(s', a') - \hat{\mathcal{R}}_\rho(s') \hat{Q}^\pi(s', a')\right]\right]\right|$$

$$(10)$$

Consider the following identity:

$$\mathcal{R}_\rho(s') Q^\pi(s', a') - \hat{\mathcal{R}}_\rho(s') \hat{Q}^\pi(s', a') = \mathcal{R}_\rho(s') Q^\pi(s', a') - \mathcal{R}_\rho(s') \hat{Q}^\pi(s', a')$$

$$+ \mathcal{R}_\rho(s') \hat{Q}^\pi(s', a') - \hat{\mathcal{R}}_\rho(s') \hat{Q}^\pi(s', a')$$

$$= \mathcal{R}_\rho(s') \left(Q^\pi(s', a') - \hat{Q}^\pi(s', a')\right)$$

$$+ \left(\mathcal{R}_\rho(s') - \hat{\mathcal{R}}_\rho(s')\right) \hat{Q}(s', a') \quad (11)$$

Plugging Eq 11 in Eq 10, we get:

$$\left|Q^\pi(s, a) - \hat{Q}^\pi(s, a)\right| = \left|\mathbb{E}_{s' \sim \mathcal{P}(\cdot|s, a)} \left[\left(\mathcal{R}_\rho(s') - \hat{\mathcal{R}}_\rho(s')\right) \left(r(s, a) + \gamma \mathbb{E}_{a' \sim \pi(\cdot|s')} [\hat{Q}^\pi(s', a')] - \frac{R_{min} - \epsilon}{1 - \gamma}\right)\right.\right.$$

$$\left.\left. + \gamma \mathcal{R}_\rho(s') \mathbb{E}_{a' \sim \pi(\cdot|s)} \left[Q^\pi(s', a') - \hat{Q}^\pi(s', a')\right]\right]\right|$$

Taking the modulus inside the expectation and using the triangle inequality, we get:

$$\left|Q^\pi(s, a) - \hat{Q}^\pi(s, a)\right| \leq \mathbb{E}_{s' \sim \mathcal{P}(\cdot|s, a)} \left[\left|\mathcal{R}_\rho(s') - \hat{\mathcal{R}}_\rho(s')\right| \left|r(s, a) + \gamma \mathbb{E}_{a' \sim \pi(\cdot|s')} [\hat{Q}^\pi(s', a')] - \frac{R_{min} - \epsilon}{1 - \gamma}\right|\right.$$

$$\left. + \gamma |\mathcal{R}_\rho(s')| \mathbb{E}_{a' \sim \pi(\cdot|s)} \left|Q^\pi(s', a') - \hat{Q}^\pi(s', a')\right|\right]$$

Using the following inequalities: $r(s,a) + \gamma \mathbb{E}_{a' \sim \pi(\cdot|s')}[\hat{Q}^\pi(s',a')] \leq R_{\max}/(1-\gamma)$ as the maximum environment reward is $R_{\max}$, $\left|\hat{\mathcal{R}}_\rho(s')\right| \leq 1$ as $\hat{\mathcal{R}}_\rho \in [0,1]$ and the assumption $\|\mathcal{R}_\rho - \hat{\mathcal{R}}_\rho\|_\infty \leq \delta$, we can complete the proof:

$$
\begin{aligned}
\left|Q^\pi(s,a) - \hat{Q}^\pi(s,a)\right| &\leq \delta \frac{R_{\max} - R_{\min} + \epsilon}{1-\gamma} + \gamma \mathbb{E}_{s' \sim \mathcal{P}(\cdot|s,a), a' \sim \pi(\cdot|s)} \left|Q^\pi(s',a') - \hat{Q}^\pi(s',a')\right| \\
&\leq \delta \frac{R_{\max} - R_{\min} + \epsilon}{1-\gamma} + \gamma \mathbb{E}_{s' \sim \mathcal{P}(\cdot|s,a), a' \sim \pi(\cdot|s)} \left[\delta \frac{R_{\max} - R_{\min} + \epsilon}{1-\gamma} + \gamma \mathbb{E} \dots\right] \\
&\leq \delta \frac{R_{\max} - R_{\min} + \epsilon}{(1-\gamma)^2}
\end{aligned}
$$

$\square$

Lemma A.4 gives us a bound on the difference between the true $Q$-values and $Q$-values computed using the reversibility estimator $\hat{\mathcal{R}}_\rho$ for any policy $\pi$. The bound and the proof also suggest that closer $\hat{\mathcal{R}}_\rho(s)$ is to $\mathcal{R}_\rho$, closer the estimated $Q$-values are to true ones.

Similar to the empirical Bellman backup, we can define the empirical Bellman optimality operator:

$$
\hat{\mathcal{B}}^* Q(s,a) = \mathbb{E}_{s' \sim \mathcal{P}(\cdot|s,a)} \left[\hat{\mathcal{R}}_\rho(s') \left(r(s,a) + \gamma \max_{a'} Q(s',a')\right) + \left(1 - \hat{\mathcal{R}}_\rho(s')\right) \frac{R_{\min} - \epsilon}{1-\gamma}\right] \quad (12)
$$

The empirical Bellman optimality operator is also a contraction, following a proof similar as that for empirical Bellman backup. Let $\hat{Q}^*$ denote the fixed point of Bellman optimality operator, that is $\hat{B}^* \hat{Q}^* = \hat{Q}^*$, and let $\hat{\pi}^*$ denote the greedy policy with respect to $\hat{Q}^*$. As a final result,

**Theorem 5.1.** *Let $\pi^*$ denote the optimal policy and $Q^*$ denote the corresponding optimal Q-value function. Let $\hat{\pi}^*$ denote the optimal policy returned by empirical Bellman optimality operator $\hat{\mathcal{B}}^*$. Assuming $\|\mathcal{R}_\rho - \hat{\mathcal{R}}_\rho\|_\infty \leq \delta$,*

$$
Q^{\hat{\pi}^*}(s,a) \geq Q^*(s,a) - \frac{2\delta (R_{\max} - R_{\min} + \epsilon)}{(1-\gamma)^2}
$$

*for all $(s,a) \in \mathcal{S} \times \mathcal{A}$.*

*Proof.* For clarity of notation, we will use $Q(\pi)$ denote $Q^\pi(s,a)$ and $\hat{Q}(\pi)$ denote $\hat{Q}^\pi(s,a)$ for a policy $\pi$. Now,

$$
Q^*(s,a) - Q^{\hat{\pi}^*}(s,a) = \left(Q(\pi^*) - \hat{Q}(\pi^*)\right) + \left(\hat{Q}(\pi^*) - \hat{Q}(\hat{\pi}^*)\right) + \left(\hat{Q}(\hat{\pi}^*) - Q(\hat{\pi}^*)\right)
$$

Using the fact $\hat{\pi}^*$ is the optimal policy with respect to $\hat{\mathcal{B}}^*$, we have $\hat{Q}(\pi^*) - \hat{Q}(\hat{\pi}^*) \leq 0$. This implies that:

$$
Q^*(s,a) - Q^{\hat{\pi}^*}(s,a) \leq \left(Q(\pi^*) - \hat{Q}(\pi^*)\right) + \left(\hat{Q}(\hat{\pi}^*) - Q(\hat{\pi}^*)\right) \quad (13)
$$

Using lemma A.4, we have

$$
\begin{aligned}
\left(Q(\pi^*) - \hat{Q}(\pi^*)\right) &\leq \frac{\delta (R_{\max} - R_{\min} + \epsilon)}{(1-\gamma)^2} \\
\left(\hat{Q}(\hat{\pi}^*) - Q(\hat{\pi}^*)\right) &\leq \frac{\delta (R_{\max} - R_{\min} + \epsilon)}{(1-\gamma)^2}
\end{aligned}
$$

Plugging these in Eq 13, we get

$$
Q^*(s,a) - Q^{\hat{\pi}^*}(s,a) \leq \frac{2\delta (R_{\max} - R_{\min} + \epsilon)}{(1-\gamma)^2}
$$

Rearranging the above bound gives us the statement in the theorem. $\square$

Theorem 5.1 gives us the assurance that as long as the estimator $\hat{\mathcal{R}}_\rho$ is close to $\mathcal{R}_\rho$, the $Q$-values of the greedy policy obtained by value iteration using $\hat{\mathcal{B}}^*$, i.e. $\hat{\pi}^*$ will not be much worse than the optimal $Q$-values. The above result also suggests choosing a smaller $\epsilon$ for a smaller gap in performance.

## A.2  Detailed Pseudocode

In Algorithms 3 and 4, we provide pseudo-code for the episodic and non-episodic variants of our method PAINT. We build the non-episodic variant upon the MEDAL algorithm [36], which introduces a backward policy whose objective is to match the state distribution of the forward demonstrations (summarized in Section 4).

---

**Algorithm 3:** PAINT (Episodic)

---

**optional:** forward demonstrations $\mathcal{N}$
**initialize:** $\pi, Q, \mathcal{D}$; // forward agent parameters
**initialize** $\hat{\mathcal{R}}_\rho, \mathcal{D}_\rho$; // reversibility classifier and dataset of labels
// add demonstrations to replay buffer
$\mathcal{D} \leftarrow \mathcal{D} \cup \mathcal{N}$;
**while** *not done* **do**
    $s \sim \rho_0$ // reset environment
    $\mathcal{D}_{\text{new}} \leftarrow \mathcal{D}_{\text{new}} \cup \{s\}$;
    aborted $\leftarrow$ False;
    **for** $t = 1, 2, \ldots, H$ **do**
        **if** *not aborted* **then**
            $a \sim \pi(\cdot \mid s)$;
            update $\pi, Q$; // Eq 9
        **else**
            $a \sim \text{unif}(\mathcal{A})$;
        $s' \sim \mathcal{P}(\cdot \mid s, a), r \leftarrow r(s, a)$;
        $\mathcal{D} \leftarrow \mathcal{D} \cup \{(s, a, s', r)\}$;
        **if** *not aborted and* $\hat{R}_\rho(s) \leq 0.5$ **then**
            aborted $\leftarrow$ True;
        $\mathcal{D}_{\text{new}} \leftarrow \mathcal{D}_{\text{new}} \cup \{s'\}$;
        $s \leftarrow s'$;
    // query reversibility labels for newly collected states via Alg. 1
    label $\mathcal{D}_{\text{new}}$;
    $\mathcal{D}_\rho \leftarrow \mathcal{D}_\rho \cup \mathcal{D}_{\text{new}}$;
    $\mathcal{D}_{\text{new}} \leftarrow \emptyset$;
    // train classifier on all labeled data, new and old
    update $\hat{\mathcal{R}}_\rho$;

---

## A.3  Environment Details

In this section, we provide details of each of the four irreversible environments, which are visualized in Fig. 4 and Fig. 5.

**Maze**. In this 2-D continuous-control environment, the agent is a point mass particle that starts in the top left corner of the maze and must reach the bottom right corner. Throughout the environment, there are trenches (marked in black) that the agent must avoid. Entering them is irreversible: the agent can roam freely within the trench but cannot leave it without an environment reset. The agent is placed back at the top left corner upon a reset, which is provided every 500 time-steps. The agent's state space consists of its $xy$-position, and two control inputs correspond to the change applied to its $xy$-position. The reward function is defined as $r_t = \mathbb{1}(\|s_t - g\|_2 < 0.1)$, where $g$ is the goal position. We provide 10 demonstrations of the task to the agent at the beginning of training.

**Tabletop Organization**. This environment modifies the Tabletop Organization task from Sharma et al. [35]. The agent's objective is to grasp the mug and move it to one of the four specified goal positions. Grasping the mug and dropping it off beyond the red boundary is irreversible, i.e., the agent can no longer re-grasp the mug. Upon a reset, which is provided every 200K time-steps or when requested by the agent, the agent is placed back at the center of the table and the mug is placed on its right, just within the red boundary (see Fig. 4). The agent's state space consists of the its own $xy$-position, the mug's $xy$-position, an indicator of whether the mug is grasped, the goal position of the mug, and finally, the goal position of the agent after putting the mug down at its goal. There are three control inputs, which apply changes to the agent's $xy$-position and toggle between grasping, if

**Algorithm 4:** PAINT with MEDAL [36] (Non-episodic)

---

**input:** forward demonstrations $\mathcal{N}_f$;
**optional:** backward demonstrations $\mathcal{N}_b$;
**initialize:** $\pi_f, Q^f, \mathcal{D}_f$; // forward agent parameters
**initialize:** $\pi_b, Q^b, \mathcal{D}_b$; // backward agent parameters
**initialize** $\hat{\mathcal{R}}_\rho, \mathcal{D}_\rho$; // reversibility classifier and dataset of labels
**initialize** $C(s)$; // state-space discriminator for backward policy
// add demonstrations to replay buffer
$\mathcal{D}_f \leftarrow \mathcal{D}_f \cup \mathcal{N}_f$;
$\mathcal{D}_b \leftarrow \mathcal{D}_b \cup \mathcal{N}_b$;
$\mathcal{D}_{\text{new}} \leftarrow \emptyset$ ;
**while** *not done* **do**
    $s \sim \rho_0$; // reset environment
    $\mathcal{D}_{\text{new}} \leftarrow \mathcal{D}_{\text{new}} \cup \{s\}$;
    // continue till the reversibility classifier detects an irreversible state
    **while** $\hat{\mathcal{R}}_\rho(s) > 0.5$ **do**
        // run forward policy for a fixed number of steps, switch to backward policy
        **if** *forward* **then**
            $a \sim \pi_f(\cdot \mid s)$;
            $s' \sim \mathcal{P}(\cdot \mid s, a), r \leftarrow r(s, a)$;
            $\mathcal{D}_f \leftarrow \mathcal{D}_f \cup \{(s, a, s', r\}$;
            update $\pi_f, Q^f$; // Eq 9
        **else**
            $a \sim \pi_b(\cdot \mid s)$;
            $s' \sim \mathcal{P}(\cdot \mid s, a), r \leftarrow -\log(1 - C(s'))$;
            $\mathcal{D}_b \leftarrow \mathcal{D}_b \cup \{(s, a, s', r)\}$;
            update $\pi_b, Q^b$; // Eq 9
        // train disriminator every $K$ steps
        **if** *train-discriminator* **then**
            // sample a batch of positives $S_p$ from the forward demos $\mathcal{N}_f$, and a batch of negatives $S_n$ from backward replay buffer $\mathcal{D}_b$
            $S_p \sim \mathcal{N}_f, S_n \sim \mathcal{D}_b$;
            update $C$ on $S_p \cup S_n$;
        $\mathcal{D}_{\text{new}} \leftarrow \mathcal{D}_{\text{new}} \cup \{s'\}$;
        $s \leftarrow s'$;
    // optionally explore environment
    **for** *explore steps* **do**
        $a \sim \text{unif}(\mathcal{A}), s' \sim \mathcal{P}(\cdot \mid s, a), r \leftarrow r(s, a)$;
        update $\mathcal{D}_f, \mathcal{D}_b$; // use $C(s)$ for the reward labels in $\mathcal{D}_b$
        $\mathcal{D}_{\text{new}} \leftarrow \mathcal{D}_{\text{new}} \cup \{s'\}$;
        $s \leftarrow s'$;
    // query reversibility labels for newly collected states via Alg 1
    label $\mathcal{D}_{\text{new}}$;
    $\mathcal{D}_\rho \leftarrow \mathcal{D}_\rho \cup \mathcal{D}_{\text{new}}$;
    $\mathcal{D}_{\text{new}} \leftarrow \emptyset$;
    // train classifier on all labeled data, new and old
    update $\hat{\mathcal{R}}_\rho$ on $\mathcal{D}_\rho$;

---

the object is nearby (i.e., within a distance of $0.4$), and releasing, if the object is currently grasped. The agent's reward function is $r_t = \mathbb{1}(\|s_t - g\|_2 < 0.1)$, i.e., both the agent's $xy$-position and the mug's $xy$-position must be close to their targets. We provide 50 forward demonstrations, 50 backward demonstrations, and 1000 examples of (randomly generated) irreversible states to the agent at the beginning of training.

**Peg Insertion**. This environment modifies the Peg Insertion task from Sharma et al. [35]. The objective of this task is to grasp and insert the peg into the hole in the box. We modified the table so that the raised edges that stop the peg from rolling off the table are removed and the table is significantly narrower. Hence, the peg may fall off the table, which cannot be reversed by the robot. Instead, when the environment is reset, which automatically occurs every 100K time-steps or when requested by the agent, the peg is placed back at one of 15 possible initial positions on the table. The agent's state space consists of the robot's $xyz$-position, the distance between the robot's gripper fingers, and the object's $xyz$-position. The agent's action consists of 3D end-effector control and normalized gripper torque. Let $s_t^{\text{peg}}$ represent the state of the peg and $g^{\text{peg}}$ be its goal state, then the reward function is $r_t = \mathbb{1}(\|s_t^{\text{peg}} - g^{\text{peg}}\|_2 < 0.05)$. We provide the agent with 12 forward demonstrations and 12 backward demonstrations.

**Half-Cheetah**. We design this environment based on the version from Brockman et al. [9]. In particular, the agent must run at one of six target velocities $\{3, 4, 5, 6, 7, 8\}$, specified to the agent. Every 500 time-steps, the target velocity switches to a different value selected at random. The agent's actions, which correspond to torque control of the cheetah's six joints, are scaled up by a factor of 5. When the agent is flipped onto its back (i.e., its normalized orientation is greater than $2\pi/3$), we label these states irreversible. When the agent is reset, which only occurs when requested, the agent is placed upright again at angle of 0. The agent's observation space consists of the velocity of the agent's center of mass, angular velocity of each of its six joints, and the target velocity. Let $v$ be the velocity of the agent, then the reward function, which is normalized to be between 0 and 1, is $r_t = 0.95 * (8 - v)/8 + 0.05 * (6 - \|a_t\|_2^2)/6$. There are no demonstrations for this task.

### A.4 Implementation Details

Below, we provide implementation details of our algorithm PAINT and the baselines. Every algorithm, including ours, has the following components.

*Forward policy network*. The agent's forward policy is represented by an MLP with 2 fully-connected layers of size 256 in all experimental domains, trained with the Soft Actor-Critic (SAC) [18] algorithm.

*Forward critic network*. The agent's forward critic is represented by an MLP with 2 fully-connected layers of size 256 in all experimental domains, trained with the Soft Actor-Critic (SAC) [18] algorithm.

*Balanced batches*. In the Tabletop Organization and Peg Insertion tasks, the agent's forward policy and critic networks are trained with batches that consist of demonstration tuples and of tuples sampled from the agent's online replay buffer. We control the ratio $p$ of demonstration tuples to online tuples with a linearly decaying schedule,

$$p_t = \begin{cases} \frac{(p_T - p_0)}{T} t + p_0 & t < T \\ p_T & T \geq t. \end{cases}$$

In the Tabletop Organization and Peg Insertion tasks, $p_0 = 0.5$, $p_T = 0.1$, and $T = 500K$. We do not train with balanced batches in the Maze task, and simply populate the online replay buffer with the demonstrations at the beginning of training.

**Episodic RL (Soft Actor-Critic)** [18]. In addition to the policy and critic networks trained with balanced batches, the episodic RL comparison requests for resets every $H'$ time-steps in the Tabletop Organization ($H' = 2000$), Peg Insertion ($H' = 1000$), and Half-Cheetah ($H' = 2000$) tasks.

**Safe Model-Based Policy Optimization (SMBPO)** [41]. This comparison trains the forward critic with the modified Bellman update $\mathcal{B}^\pi Q(s, a)$ defined in Eqn. 4, where $\epsilon = 0$ in the Maze and Half-Cheetah tasks and $\epsilon = -0.1$ in the Tabletop Organization and Peg Insertion tasks.

**Safety Q-functions for RL (SQRL)** [37]. This comparison trains an additional safety critic $Q_{\text{safe}}^\pi$, which estimates the future probability of entering an irreversible state, and the policy is updated with the following augmented objective

$$J_{\text{safe}}^\pi(\nu) = J^\pi + \mathbb{E}_{s \sim \mathcal{D}, a \sim \pi(\cdot|s)} \left[ \nu \left( \epsilon_{\text{safe}}, Q_{\text{safe}}^\pi(s, a) \right) \right],$$

where $\nu$ is the Lagrange multiplier for the safety constraint and is updated via dual gradient descent. We only evaluate this comparison in the Maze task, where $\epsilon_{\text{safe}} = 10$.

### A.4.1  Forward-Backward Algorithms

In the forward-backward setups, we train a backward policy and critic in addition to their forward counterparts. The details of the backward components that are shared across all methods are described below.

*Backward policy network.* The agent's backward policy is represented by an MLP with 2 fully-connected layers of size 256 in all experimental domains, trained with the SAC algorithm.

*Backward critic network.* The agent's backward critic is represented by an MLP with 2 fully-connected layers of size 256 in all experimental domains, trained with the SAC algorithm.

*Backward balanced batches.* The agent's backward policy and critic networks are trained with batches that consist of demonstration tuples and of tuples sampled from the agent's online replay buffer. We control the ratio $p$ of demonstration tuples to online tuples with a linearly decaying schedule,

$$p_t = \begin{cases} \frac{(p_T - p_0)}{T} t + p_0 & t < T \\ p_T & T \geq t. \end{cases}$$

In the Tabletop Organization and Peg Insertion tasks, $p_0 = 0.5$, $p_T = 0.1$, and $T = 500\text{K}$.

**Leave No Trace (LNT)** [11]. This comparison additionally trains backward policy and critic networks, whose reward function is the sparse indicator of whether the current state is within some threshold of the initial state. The thresholds are the same as those used for the forward reward functions defined in A.3.

*Policy switching.* Leave No Trace switches from the forward to backward policy if the backward critic's $Q$-value is lower than $\epsilon_{\text{LNT}}$ or after 300 time-steps, and switches from the backward to forward policy after 300 time-steps. In Tabletop, $\epsilon_{\text{LNT}} = 0.1$, and in Peg Insertion, $\epsilon_{\text{LNT}} = 0.005$.

*Termination condition.* Leave No Trace additionally requests a reset if, after 300 time-steps, the backward policy fails to bring the environment within a distance of 0.1 of the initial state.

**Matching Expert Distributions for Autonomous Learning (MEDAL)** [36]. Like Leave No Trace, MEDAL trains a backward policy and critic. However, instead of returning to the initial state, the backward reward function is whether the current state matches the distribution of demonstration states, formally defined in Eqn. 1.

*MEDAL classifier.* The classifier $C$ in Eqn. 1 is represented by an MLP with 1 FC layer of size 128.

*Policy switching.* The algorithm switches policies (i.e., from forward to backward and from backward to forward) after every 300 time-steps.

**Reversibility-Aware Exploration (RAE)** [15]. RAE trains a self-supervised reversibility estimator, specifically to predict whether a state transition $(s, \tilde{s})$ is more likely than the reverse transition $(\tilde{s}, s)$. RAE generates data for the binary classifier with a windowed approach. For every state trajectory $(s_{t:t+w})$ of length $w$ collected by the agent, all state pairs $(s_i, s_j)$, where $i < j$, are labeled *positive*, and all pairs $(s_j, s_i)$, where $i < j$, are labeled *negative*. For all experimental tasks, we use a window size of $w = 10$ time-steps. With this estimator, the forward critic is trained with the modified Bellman update $\hat{\mathcal{B}}^\pi Q(s, a)$, where $\epsilon = 0$ in the Maze and Half-Cheetah tasks and $\epsilon = -0.1$ in the Tabletop Organization and Peg Insertion tasks.

*Reversibility classifier.* The classifier $\hat{\mathcal{R}}_\rho$ is represented by an MLP with 1 FC layer of size 128.

*Termination condition.* In Maze and Half-Cheetah, the termination condition is $\hat{\mathcal{R}}_\rho > 0.5$. In Tabletop Organization and Peg Insertion, the condition is $\hat{\mathcal{R}}_\rho > 0.8$.

*Exploration.* We augment RAE with uniform-random exploration after the termination condition is met as proposed in our method. In the Maze environment, the agent takes uniform-random actions for the rest of the episode (of length 500). In Tabletop Organization and Peg Insertion, $N_{\text{explore}} = 300$ time-steps. In Half-Cheetah, $N_{\text{explore}} = 500$ time-steps.

In the Tabletop Organization and Peg Insertion tasks, we train an additional backward policy and critic, whose reward functions are defined in terms of the MEDAL classifier. The backward critic is also trained with the modified $\hat{\mathcal{B}}^\pi Q(s, a)$, with the same hyperparameters as the forward critic.

*MEDAL classifier.* The classifier $C$ in Eqn. 1 is represented by an MLP with 1 FC layer of size 128.

*Policy switching.* The algorithm switches policies (i.e., from forward to backward and from backward to forward) after every 300 time-steps.

**PAINT (Ours).** PAINT trains a reversibility classifier $\hat{\mathcal{R}}_\rho$ and checks whether the current state is estimated to be irreversible. If it is estimated to be irreversible, the agent takes uniform-random actions for $H_{\text{explore}}$ time-steps and requests a reset afterward. The forward critic is also trained with the modified Bellman update $\hat{\mathcal{B}}^\pi Q(s, a)$, where $\epsilon = 0$ in the Maze and Half-Cheetah tasks and $\epsilon = -0.1$ in the Tabletop Organization and Peg Insertion tasks.

*Reversibility classifier.* The classifier $\hat{\mathcal{R}}_\rho$ is represented by an MLP with 1 FC layer of size 128.

*Termination condition.* In all tasks, the termination condition is $\hat{\mathcal{R}}_\rho > 0.5$.

*Exploration.* In the Maze environment, the agent takes uniform-random actions for the rest of the episode (of length 500). In Tabletop Organization and Peg Insertion, $N_{\text{explore}} = 300$ time-steps. In Half-Cheetah, $N_{\text{explore}} = 500$ time-steps.

In the Tabletop Organization and Peg Insertion tasks, we train an additional backward policy and critic. The details for the backward policy and critic are the same as in RAE.

### A.4.2 Codebase

We have publicly released our code at this GitHub repo: `https://github.com/tajwarfahim/proactive_interventions`. Our codebase builds on top of codebases from Yarats et al. [48, 49] and Sharma et al. [35].

### A.5 Discussion on Comparisons

| Method | Forward-Backward | Requires Reversibility Labels? | Intervention Rule |
|---|---|---|---|
| Episodic RL (SAC) [18] | No | No | N/A |
| SMBPO [41] | No | Yes | N/A |
| SMBPO w. Oracle Term | No | Yes | Oracle |
| SQRL [37] | No | Yes | N/A |
| LNT [11] | Yes | No | $s \notin \text{supp}(\rho_0)$ |
| MEDAL [36] | Yes | No | N/A |
| RAE [15] | Yes | No | $\hat{\mathcal{R}}_{\text{RAE}}(s) > p$ |
| PAINT (Ours) | Yes | Yes | $\hat{\mathcal{R}}_\rho(s) < 0.5$ |

Table 1: Summary of assumptions for each method.

The algorithms we compare to make varying assumptions, which we summarize in Table 1. In particular, SMBPO [41] and SQRL [37] are safe RL methods, which we adapt to the setting in this work. They use reversibility labels as safety labels, but lack a backward policy to reset the agent and an intervention rule. We therefore only study these comparisons in the episodic setting. In the continuing setup, we equip SMBPO with an oracle intervention rule, which calls for a reset when the agent enters an irreversible state. The next comparisons LNT [11] and MEDAL [36] are explicitly designed for the autonomous RL setting: they train forward and backward policies, which take alternate turns controlling the agent. They however do not use reversibility labels. Instead, LNT requires a different assumption, checking whether the agent has returned back to the support of the initial state distribution, while MEDAL has no defined intervention rule. Finally, we adapt RAE [15], which is not originally designed for autonomous RL, by introducing a backward policy and designing an intervention rule based on the RAE reversibility classifier. Notably, RAE trains its classifier with self-supervised labels, and so does not require any explicit reversibility labels provided by an expert.

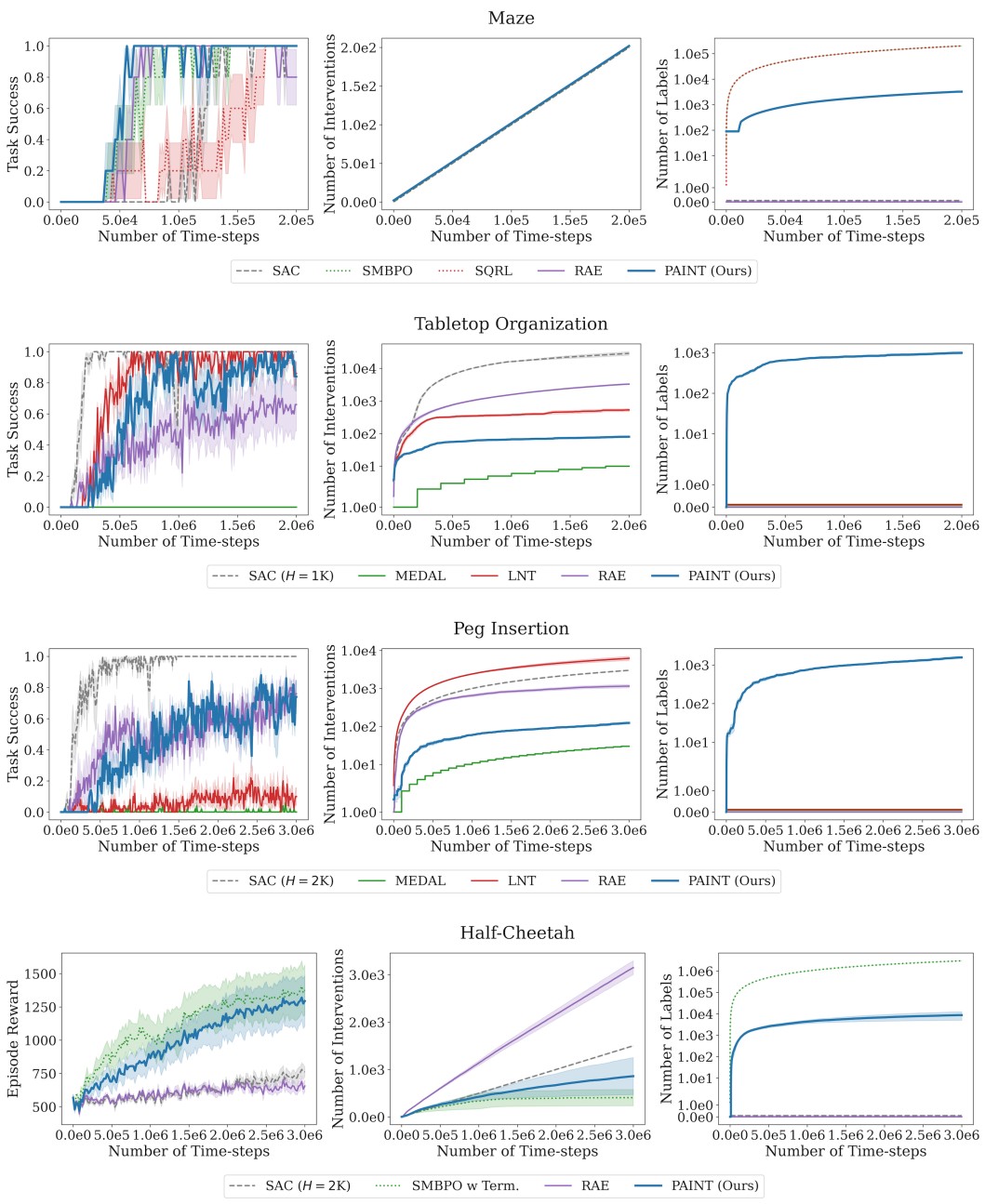

Figure 9: *(left)* Task success versus time. *(middle)* Number of interventions versus time. *(right)* Number of queried labels versus time.

## A.6 Additional Experimental Results

In this section, we present additional plots to accompany Section 6.2 and additional ablations to accompany Section 6.3.

### A.6.1 Additional Plots

In Fig. 9, we plot the task success, number of reset interventions, and number of reversibility labels versus the number of time-steps in each of the four tasks.

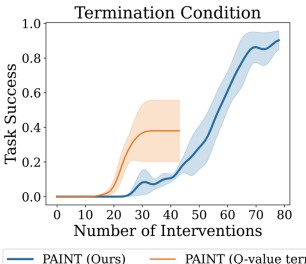
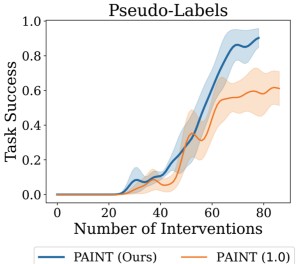

Figure 10: (*left*) We define a termination condition based on the $Q$-values, which learns with fewer resets but also achieves significantly lower final performance. (*right*) For states collected during a trial, their reversibility labels are still unknown. When training the agent with Eqn. 9, we use predictions from the reversibility classifier as pseudo-labels. We compare to labeling the unlabeled states one, i.e., treating them as reversible.

### A.6.2  Ablations and Sensitivity Analysis

**Termination conditions**. To evaluate the importance of the reversibility classifier, we study an alternative choice for the termination condition, one based on the $Q$-value function. Intuitively, the values for irreversible states will be low, as there is no path that leads to the goal from these states. We define the value-based termination condition as $V^\pi(s) < \epsilon$ where $\epsilon$ is the threshold. We approximate the value $V^\pi(s) = \mathbb{E}_\pi[Q^\pi(s,a)]$ with the trained $Q$-value function evaluated at $N = 10$ policy actions. We plot the task success in the Tabletop Manipulation task in Fig. 10 (left). After 1M time-steps, the $Q$-value-based termination condition requires fewer interventions, approximately half of the interventions needed by PAINT, but does not converge to the same final performance as PAINT with the reversibility classifier. Critically, PAINT with $Q$-value termination also trains a reversibility classifier to generate pseudo-labels for unlabeled states.

**Pseudo-labels for unlabeled states**. Currently, when updating the agent with Eqn. 9, our method uses the predictions from the stuck classifier as pseudo-labels for the unlabeled states collected during a trial. An alternative choice is labeling them one, i.e., treating them as reversible. We evaluate this choice in the Tabletop Manipulation task, and plot the task success in Fig. 10 (right). After 1M time-steps, the number of reset interventions requested by both methods are similar. However, our method succeeds at the task almost $100\%$ of the time, while the agent trained with pseudo-labels of one only achieves success of around $60\%$.

**Reversibility classifier threshold.** The threshold that determines when to request an intervention can affect the performance of PAINT. Setting the threshold too low can result in the agent being stuck in irreversible states for a long time before calling for a reset while too high a threshold may trigger the reset too many times. To understand its sensitivity, we evaluated PAINT on the following threshold values, $\{0.1, 0.3, 0.5, 0.7, 0.9\}$, in the non-episodic tasks. The results, visualized in Fig. 11, indicate that PAINT with larger values like $0.9$ can perform worse, but across all tasks, a threshold value of $0.5$ yields high success rates while requiring few reset interventions.

**Reducing label queries with classifier confidence.** Rather than querying labels at every iteration of our binary search procedure, we study a variant of PAINT where we query labels from the expert only when there is uncertainty in the prediction. Specifically on the Maze environment, we study two forms of uncertainty: (a) the confidence of the classifier given by its output probability (predictions close to $0.5$ are uncertain) and (b) epistemic uncertainty captured by an ensemble of classifiers. Only when the predictor is uncertain, i.e., $|\hat{\mathcal{R}}_\rho(s) - 0.5| < p$ in (a) and $\mathrm{STD}(\hat{\mathcal{R}}_\rho^i(s)) > p$ in (b), do we need to query the reversibility label. Otherwise, we can use the classifier's prediction in place of querying, which can further reduce the number of labels required by PAINT.

In Fig. 12 (top), we visualize the results of PAINT that queries based on the confidence of the classifier, when $|\hat{\mathcal{R}}_\rho(s) - 0.5| < p$ for $p \in \{0.1, 0.4\}$, and we find that PAINT can achieve similar success rates even when it only queries labels for states with low-confidence predictions. In particular, PAINT with confidence-based querying requires fewer than $100$ labels compared to the $3K$ labels required by standard PAINT. In Fig. 12 (bottom), we report the results for PAINT with querying based on the uncertainty of an ensemble of classifiers, when $\mathrm{STD}(\hat{\mathcal{R}}_\rho^i(s)) > p$ for $p \in \{0.01, 0.001, 0.0001\}$. Here, we find that a threshold of $0.0001$ achieves high success rates while only requiring around $750$

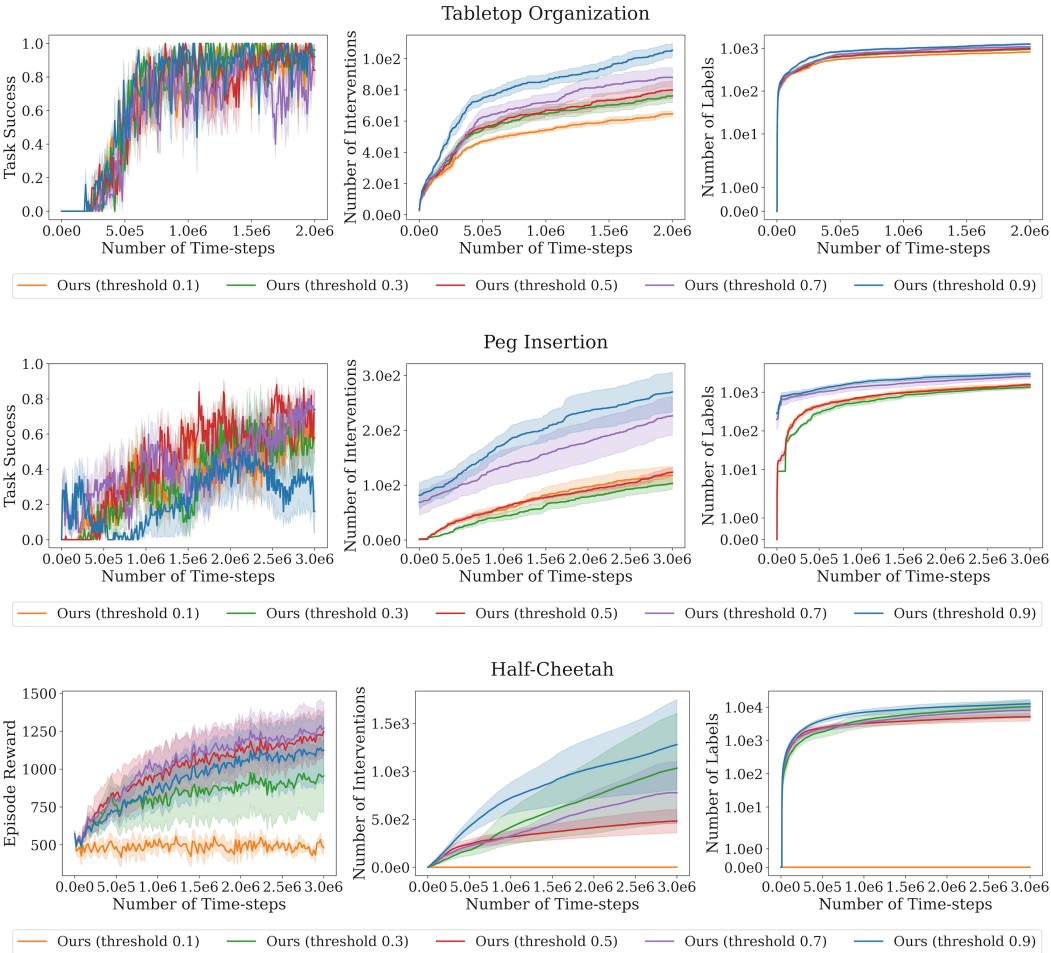

Figure 11: Task success (*left*), number of interventions (*middle*), and number of labels (*right*) versus time for different threshold values.

labels. These are promising improvements, and future work can extend them to more domains and more sophisticated querying strategies.

**Noisy reversibility labels**. Thus far, our method has assumed the reversibility labels are noiseless. However, these labels in most practical settings come from human supervisors, who can inadvertently introduce noise into the labeling process. Hence, we simulate noisy reversibility labels in the maze environment, and design a *robust* variant of our labeling scheme to account for possible noise in the labels. In the robust variant, in addition to querying the label for a state $s$, we also query the neighboring states, i.e., the sequence of $N$ states centered at $s$, and take the majority as the label for $s$.

We design two different noisy scenarios with (a) false positive labels and (b) false negative labels in the Maze environment. In (a), the trench regions can produce noisy labels. With probability $0.2$, the label is $1$, incorrectly labeling them as *reversible*. In (b), the regions of the state space that neighbor the trenches can produce noisy labels. That is, with probability $0.2$, the label is $0$, incorrectly labeling them as *irreversible*. In Fig. 13 (left and middle), we see that using a window size of $10$, PAINT succeeds $60\%$ of the time under false positive labels and $80\%$ under false negative labels, while smaller window sizes perform worse. The robust variant of PAINT can therefore tolerate some degree of label noise and only increase the number of reversibility labels required by a constant factor, thereby maintaining the same label complexity of $\mathcal{O}\left(N \log |\tau|_{\max}\right)$ as before.

**Noisy transitions**. We also incorporate noise to the transition dynamics to further evaluate the robustness of PAINT. In the Maze environment, we add Gaussian noise of width $\sigma$ to the action before transitioning to the next state. The action space is $[-1.0, 1.0]^2$, and we evaluate on $\sigma$ values

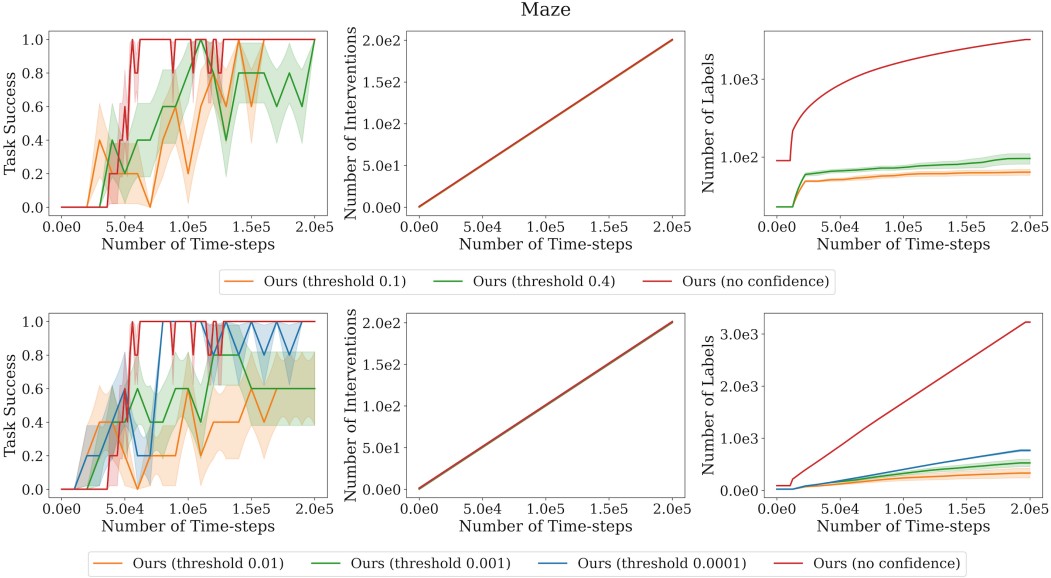

Figure 12: (*top*) Querying based on the confidence of the classifier, i.e., when $|\hat{\mathcal{R}}_\rho(s) - 0.5| < p$ for $p \in \{0.1, 0.4\}$. (*bottom*) Querying based on the uncertainty of an ensemble of classifiers, i.e., when $\text{STD}(\hat{\mathcal{R}}_\rho^i(s)) > p$ for $p \in \{0.01, 0.001, 0.0001\}$.

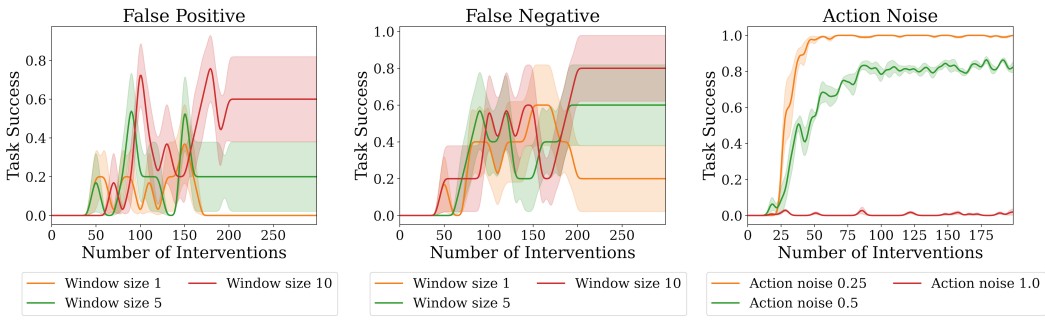

Figure 13: A robust variant of PAINT which queries neighboring states inside a window size of $1, 5$, and $10$ with potential false positive labels (*left*) and false negative labels (*middle*). (*right*) PAINT under varying noise added to the transitions.

of $0.25, 0.5$, and $1.0$. In Fig. 13 (right), we find that PAINT is robust up to a noise level of $0.5$. Unsurprisingly, when the noise is larger than the actions themselves, the agent fails to solve the task.