# OpenReview forum: "When to Ask for Help: Proactive Interventions in Autonomous Reinforcement Learning"
_NeurIPS.cc/2022/Conference — NeurIPS 2022 Accept_

### Official Review · Reviewer_pWgT · 2022-07-07

**Rating:** 7
**Confidence:** 3
**Soundness:** 3 good
**Presentation:** 4 excellent
**Contribution:** 2 fair

**Summary:**

This paper tackles the problem of training an RL system in an environment where not all actions are reversible, and therefore the environment must be reset before further training can take place. The approach learns a classifier which labels states with a reversibility label R -> [0,1] and only requests an intervention to reset the environment when this passes a threshold. The classifier training exploits the fact that irreversibility is deterministic once an irreversible state has been reached (i.e. all future states are irreversible too). The approach is evaluated on suitably complex domains, and demonstrates that it requests fewer resets than competitor methods.

**Questions:**

1. How would system performance change if the threshold of R>=0.5 was varied.
2. Why does task performance not improve to levels of the baselines.
3. How easy or hard is it to label and the reversibility of states in control tasks like the peg in hole? Is the object observable once it falls?
4. How does uncertainty impact the problem, e.g. if an action has only some probability of leading to an irreversible state.


**Limitations:**

Some limitations are mentioned, and there are no obvious negative social implications.

**Strengths And Weaknesses:**

Overall I think the paper well written. I like that the approach is straightforward and clearly justified. It leverages existing work (MEDAL) and extends it clearly, adding a new capability. The system is evaluated on domains with well-justified irreversibility, and on these domains the results demonstrate that the proposed method reduces the number of resets requested.


On the negative side, I was disappointed to see that there was no exploration of the threshold used to trigger the resets. Setting this to 0.5 feels like a reasonable thing to do, but I imagine your results depend on this parameter in interesting ways. Similarly, the idea that the agent can execute a fixed number of exploration steps after a reset occurs seems to be thrown in rather late in the paper and not sufficiently justified or explored. I could imagine settings (e.g. maze with all trenches connected) where a single failure could allow all mapping of irreversible states - and this could be a bug or a feature.

I also would have liked to have seen more discussion on the task metrics in the evaluation. My reading of the results is that PAINT always produces better performance with fewer resets, but then plateaus and does not ask for further resets. This seems to hinder task performance. I understand that this is part of the motivation of the method (fewer resets/reversibility labels), but being unable to obtain competitive task performance seems like a major weakness, particularly if the method is terminating early.

I like the title, but I think it's misleading. In particular, there are two forms of "help": reversibility labelling and environment resetting. I think the title starts the paper by blurring these ideas, and it doesn't recover totally (even though both types of help are delivered at the same time).

---

> ### Author Response · Authors · 2022-08-02
> **Response to Reviewer pWgT**
>
> We appreciate the detailed review, questions, and suggestions for additional experiments. We answer the questions below, which we believe have improved the paper. Please let us know if there are additional concerns.
>
> > Why does task performance not improve to levels of the baselines.
>
> The performance of PAINT is comparable to or better than other autonomous RL methods. We can see this in the plots of success versus environment steps in Appendix A.5.1 (Figure 9, left), which provide an alternative visualization of the performance. On the Tabletop task, PAINT is comparable to Leave No Trace and better than other baselines, while on Peg Insertion, PAINT is comparable to RAE and better than other baselines. Critically, PAINT also requires the fewest reset interventions by a significant margin. The only methods that attain strictly better performance are methods that make oracle assumptions, either episodic resets (SAC) or oracle termination upon entering an irreversible state (SMBPO with oracle termination).
>
>
> > no exploration of the threshold used to trigger the resets
>
> > How would system performance change if the threshold of R>=0.5 was varied.
>
> Thank you for the suggestion. Indeed, the parameter affects performance – setting the threshold too low can result in the agent being stuck in irreversible states for a long time before calling for a reset while too high a threshold may trigger the reset too many times. We evaluated PAINT on the following threshold values, {0.1, 0.3, 0.5, 0.7, 0.9}, in the non-episodic tasks. The results are visualized in Figure 11 of Appendix A.6.2, and we find that the best threshold varies across tasks. For example, a threshold of 0.1 yields comparable performance with fewer resets in the Tabletop task. Across all tasks, however, a threshold value of 0.5 yields high success rates while requiring few reset interventions.
>
>
> > Is the object observable once it falls? How easy or hard is it to label … the peg in hole?
>
> > How does uncertainty impact the problem, e.g. if an action has only some probability of leading to an irreversible state.
>
> Yes, the peg is still observable by the agent after falling off the table. Labeling is easy in the peg-in-hole task, where the labeler only needs to know whether the peg has fallen. In other scenarios, labeling can be more difficult and label noise can emerge. To understand its impact, we studied a setting in which labels have (i) false positives with 20% probability and (ii) false negatives with 20% probability. We designed a robust variant of PAINT for this setting: in addition to querying the label for a state $s$, we also query the neighboring states, i.e., the sequence of $N$ states centered at $s$, and take the majority as the label for $s$. As shown in Figure 13 (left and middle), PAINT with a window size of 10 achieves success rates of around 60% and 80% in the false positive and the false negative settings, respectively.
>
> Next, we studied a setting with noisy transitions, as suggested. We added Gaussian noise of different widths to the actions in the Maze environment, and found that PAINT is robust to reasonable degrees of noise (results reported in Figure 13 (right) of Appendix A.6.2).
>
>
> > the idea that the agent can execute a fixed number of exploration steps … allow mapping of irreversible states
>
> Executing a fixed number of exploration steps allows the agent to gather more examples of irreversible states (assuming the agent switches to exploration at an irreversible state). Since exploration is autonomous and the number of exploration steps is small relative to the total number of steps between resets, the cost of such exploration is extremely low. The ablative experiment in Figure 8 (left) of Section 6.3 also highlights the importance of these exploration steps: when removed, the PAINT agent requires twice the number of resets to solve the task.
>
>
> > I like the title … and it doesn’t totally recover
>
> We agree that there are two forms of help that are considered in this work. The current title highlights the differences from prior work on autonomous RL but we are happy to consider alternative title suggestions. We also clarify the two distinct forms on Lines 26-28 in the introduction.

---

> > ### Comment · Reviewer_pWgT · 2022-08-08
> > **Thanks!**
> >
> > Thanks for your response - it helped me understand your contributions better. I note the extra results in the appendix, thanks.
> >
> > My remaining question (which doesn't require discussion here - I realise I've left this late), is which environments/tasks are really amenable to exploration after an irreversible state has been entered. E.g. when the peg drops or the cheetah flips, does exploration really provide any additional value? And does this provide another limitation? You state that your performance depends on this (free... in some sense) exploration, but how often is meaningful post-irreversilble-state exploration available in a domain where a human really is required to intervene?

---

> > > ### Comment · Reviewer_pWgT · 2022-08-08
> > > **Title**
> > >
> > > I forgot to comment on the title.  I find "When to Ask for Help" and "Proactive Interventions" just too over-suggestive, particularly the former. How about "Don't look back in anger: learning from irreversible states in auto-RL"? (well, you did ask!)

---

> > > > ### Author Response · Authors · 2022-08-08
> > > > **Response to Reviewer pWgT**
> > > >
> > > > Thanks for the response and the title suggestion!
> > > >
> > > > We'd like to clarify that the post-irreversible state exploration is primarily beneficial for the classifier - to generate more examples of irreversible states. We do not expect these states to be useful for policy learning beyond learning to avoid irreversible states.
> > > > As one example, when the peg drops from the table, the robot can still move its arm around, which generates data that helps the classifier to learn that the arm position is irrelevant to whether a state is reversible. Similarly, the half-cheetah can still move its joints when flipped over, to create additional examples of irreversible states. Without these additional examples, the classifier has very few negative examples to learn from.

---

> > > > > ### Comment · Reviewer_pWgT · 2022-08-09
> > > > > **(why does everything need a title?)**
> > > > >
> > > > > "the arm position is irrelevant to whether a state is reversible" - ah, I thought that the arm position was part of the state. This makes more sense.
> > > > >
> > > > > Thanks!

---

> ### Author Response · Authors · 2022-08-05
> **Response Follow-Up**
>
> Thank you again for your review and detailed suggestions! Please let us know if our response addresses your questions and whether there is additional clarification we can provide.

---

### Official Review · Reviewer_pQyz · 2022-07-10

**Rating:** 6
**Confidence:** 2
**Soundness:** 3 good
**Presentation:** 2 fair
**Contribution:** 2 fair

**Summary:**

The paper proposes and uses the observation that states preceding a reversible state are reversible and states succeeding an irreversible state are irreversible. Leveraging this observation, the paper proposes an approach that predicts and tries to avoid irreversible states that makes is more efficient that than traditional safe-RL methods. The proposed approach has been validated on different RL setups and shown to be better than baseline approaches.

**Questions:**

1. Why is the font size changed during the introduction of the PAINT acronym?
2. Why is it sufficient to only rely on the states preceding a reversible state and states following an irreversible state?
3. How do the safe RL approaches perform if they use the observation regarding reversible and irreversible states proposed in the present study?


**Limitations:**

Yes, the conclusion describes a couple of the limitations of the proposed method. It might be worth conjecturing how PAINT might function on shared autonomy tasks.

**Strengths And Weaknesses:**

Strengths
1. Work is related to a popular subtopic in the field.
2. Videos of experimental results are available online.
3. Decent coverage of ablation studies and limitations of current work.


Growth opportunities
1. Some thresholds, such as R_{p} appear to be arbitrary.
2. The form of the reversibility classifier is unclear.

---

> ### Author Response · Authors · 2022-08-02
> **Response to Reviewer pQyz**
>
> We thank the reviewer for their comments and highlighting opportunities for improvement. We address the concerns below, which we believe have improved the paper. Please let us know if there are further questions.
>
>
> > Some thresholds, such as R_{p} appear to be arbitrary.
>
> To understand the sensitivity of PAINT to the threshold parameter, we evaluated PAINT on the following threshold values, {0.1, 0.3, 0.5, 0.7, 0.9}, in the non-episodic tasks. The results are visualized in Figure 11 of Appendix A.6.2, and we find that the best threshold varies across tasks. For example, a threshold of 0.1 yields comparable performance with fewer resets in the Tabletop task. However, across all tasks, a threshold value of 0.5 achieves high success rates while requiring few reset interventions. The threshold 0.5 is also natural since we balance the dataset on which the binary classifier is trained.
>
>
> > Why is it sufficient to only rely on the states preceding a reversible state .. irreversible state
>
> Every state is either reversible or irreversible. When we query the label of a state, we can conclude either (a) that all states preceding the current state are reversible (if the current state is reversible) or (b) all states proceeding the given state are irreversible (if the current state is irreversible). For the other half, we continue to query labels using binary search, eventually generating the labels for all the states.
>
>
> > How do the safe RL approaches … if they use present study?
>
> Our proposed binary search scheme is not directly applicable to safe RL methods because these methods assume that the reversibility label is provided immediately when a new state is visited at each timestep. The binary search scheme queries the reversibility labels only at the end of each episode, which can be as long as tens of thousands of time-steps in some of our experiments. PAINT overcomes this by using pseudo-labels provided by our reversibility classifier until the true labels are determined at the end of the episode.
>
>
> > The form of the reversibility classifier is unclear.
>
> The implementation details can be found in Appendix A.4.1. The reversibility classifier is parameterized as a fully connected neural network with 1 hidden layer of size 128. We have also clarified this in Lines 236-237 of Section 5.4.

---

> > ### Comment · Reviewer_pQyz · 2022-08-08
> > **Follow up on safe RL approach**
> >
> > Thanks for considering the feedback, the response, and the changes made. The changes made helps with reproducibility.
> >
> > My question on the safe RL approach was whether the use of the observation that states preceding a reversible state are reversible and states succeeding an irreversible state are irreversible provide efficiency gains over the assumption regarding the reversibility label at each new state.

---

> > > ### Author Response · Authors · 2022-08-09
> > > **Response to Reviewer pQyz**
> > >
> > > To expand on our previous response, we can only run the binary search scheme at the _end_ of the episode. Hence, to apply the observation (i.e., the binary search scheme) to safe RL methods, we would need to wait for the episode to end before we can query labels. Critically, we cannot train the safe RL agent on states without labels. Hence, the agent cannot train on _any_ of the experiences from the current episode, which can be tens of thousands of time-steps in our experiments. Please let us know if this answers your question!

---

> ### Author Response · Authors · 2022-08-05
> **Response Follow-Up**
>
> Thank you again for the detailed review! Please let us know if our response addresses your questions and whether there is additional clarification we can provide.

---

### Official Review · Reviewer_EVu5 · 2022-07-12

**Rating:** 6
**Confidence:** 4
**Soundness:** 3 good
**Presentation:** 4 excellent
**Contribution:** 3 good

**Summary:**

This paper presents an approach to reduce the number of user interactions necessary to reset the underlying system in a value function based reinforcement learning task with states that the system can not reset itself. To achieve this the approach addresses 2 main issues, namely i) reducing the likelihood that the system will enter an irreversible state through a change in the reward function, ii) reducing the number of times that a user has to provide information about the state by identifying relevant candidates to find the first irreversible state.
The resulting approach can be used in the context of any value-function
based RL algorithm and the paper presents performance comparisons with other methods (although those are not exactly designed for the same problem).

**Questions:**

The querying method for label information used in this paper uses a binary search based criterion to elicit label information to build the classifier for irreversible states. Why does the certainty in the prediction not influence whether a label query is made ?

In the performance comparisons, the proposed algorithm seems to end up with lower overall success than some comparison methods although with convergence with lower interaction numbers. How does this trade off play into decisions ?


**Limitations:**

The paper discusses limitations of the approach in the last section and touches on the most important aspects. A little more discussions on the approach’s impact on other performance measures would have been useful.

**Strengths And Weaknesses:**


The paper addresses a problem in autonomous learning and human-machine  collaboration that has potential applications in a number of domains. It is well written and clearly introduces the concepts as well as the formalism where it builds on a reward function modification and a binary search-based strategy to guide the learning and the querying process.
The biggest weaknesses of the paper are in the somewhat limited amount of overall impact and in the experimental evaluation where results are compared with algorithms that were not directly designed with the same objectives in mind. It would have been good to see more discussion on the different assumptions these algorithms make.
Another issue is in Figure 7 where for the left two graphs the legend indicates 5 curves but only 4 seem to be visible.

---

> ### Author Response · Authors · 2022-08-02
> **Response to Reviewer EVu5**
>
> We thank the reviewer for their thoughtful comments. Below, we address the concerns raised, which we believe have improved the paper. Please let us know if there are further questions.
>
>
> > In performance comparisons … How does this tradeoff play into decisions?
>
> The performance of PAINT is comparable to or better than other autonomous RL methods. We can see this in the plots of success versus environment steps in Appendix A.5.1 (Figure 9, left), which provide an alternative but equivalent view of performance. On the Tabletop task, PAINT is comparable to Leave No Trace and better than other baselines, while on Peg Insertion, PAINT is comparable to RAE and better than other baselines. Critically, PAINT also requires the fewest reset interventions by a significant margin. The only methods that attain strictly better performance are methods that make oracle assumptions, either episodic resets (SAC) or oracle termination upon entering an irreversible state (SMBPO with oracle termination).
>
>
> > results are compared with algorithms that were not directly designed with the same objectives in mind … discussion on different assumptions these algorithms make
>
> Indeed, the comparisons were not directly designed for the exact setting we study. We have added a discussion in Appendix A.5 (and referenced on Lines 287-288) to clarify the assumptions made by different algorithms.
>
>
> > Why does the certainty in the prediction not influence whether a label query is made?
>
> Thank you for the suggestion. We ran an additional experiment with a variant of PAINT in which we query labels from the expert only when the predictions are uncertain. Specifically, we study two forms of uncertainty: (1) the confidence of the classifier given by its output probability (predictions close to 0.5 are uncertain) and (2) epistemic uncertainty captured by an ensemble of classifiers. In Fig. 12 of Appendix A.6.2, we visualize the results for both experiments. With appropriately chosen thresholds, we can reduce the number of labels from ~3K to ~100 with confidence-based querying and ~750 with ensemble-based querying. This is a promising improvement, and future work can extend this to more domains and more sophisticated querying strategies.
>
>
> > Another issue is in Figure 7 … 4 seem to be visible
>
> We apologize for the confusion. MEDAL receives fixed interventions every 200K steps and does not learn how to solve the task. As a result, the performance curve for MEDAL is a small segment at the bottom left corner, referenced on Lines 306-307. We have now clarified this in the caption of Figure 7.

---

> ### Author Response · Authors · 2022-08-05
> **Response Follow-Up**
>
> Thank you again for the insightful comments! Please let us know if our response addresses your questions and whether there is additional clarification we can provide.

---

### Author Response · Authors · 2022-08-02
**General Response**

We appreciate the constructive feedback from all of the reviewers. Below is a summary of changes, which we believe have significantly improved the clarity and experimental analysis of the paper.
- We evaluate performance while varying the threshold for the reversibility classifier in Fig. 11 of Appendix A.6.2 (Reviewers pQyz and pWgT)
- We query labels based on the confidence of the reversibility classifier in Fig. 12 of Appendix A.6.2. The number of labels is reduced from ~3K to ~100 while task success is preserved on the Maze environment (Reviewer EVu5)
- We analyze how noise in the reversibility labels affects performance in Fig. 13 of Appendix A.6.2, and propose a robust variant of our binary search routine for labeling trajectories under such noise (Reviewer pWgT)
- We analyze how noise in the transition dynamics affects performance in Fig. 13 of Appendix A.6.2 (Reviewer pWgT)
- We discuss prior methods and their assumptions in Appendix A.5 (Reviewer EVu5)

---

### Meta-Review · Area_Chair_8WCX · 2022-08-26

**Recommendation:** Accept
**Confidence:** Certain

**Metareview:**

The paper introduces a new problem statement for RL, i.e., how to identify irreversible states in RL that require help from a human operator. The problem statement and the algorithm are well motivated and the the reviewers also appreciated the reported experiments. The authors addressed the few concerns (ablations, comparisons) well in the rebuttal. I follow the reviewers and recommend acceptance.

**Award:**

No

---

### Decision · Program_Chairs · 2022-09-14

Accept